# FraC nanopores with adjustable diameter identify the mass of opposite-charge peptides with 44 dalton resolution

Gang Huang[1], Arnout Voet[2] & Giovanni Maglia[1]

A high throughput single-molecule method for identifying peptides and sequencing proteins based on nanopores could reduce costs and increase speeds of sequencing, allow the fabrication of portable home-diagnostic devices, and permit the characterization of low abundance proteins and heterogeneity in post-translational modifications. Here we engineer the size of Fragaceatoxin C (FraC) biological nanopore to allow the analysis of a wide range of peptide lengths. Ionic blockades through engineered nanopores distinguish a variety of peptides, including two peptides differing only by the substitution of alanine with glutamate. We also find that at pH 3.8 the depth of the peptide current blockades scales with the mass of the peptides irrespectively of the chemical composition of the analyte. Hence, this work shows that FraC nanopores allow direct readout of the mass of single peptide in solution, which is a crucial step towards the developing of a real-time and single-molecule protein sequencing device.

---

[1] Groningen Biomolecular Sciences and Biotechnology Institute, University of Groningen, 9747 AG Groningen, The Netherlands. [2] Laboratory of Biomolecular Modelling and Design, Department of Chemistry, University of Leuven, Celestijnenlaan 200G, 3001 Heverlee, Belgium. Correspondence and requests for materials should be addressed to G.M. (email: g.maglia@rug.nl)

Proteins regulate nearly all life processes. Currently, mass spectrometry is the method of choice for protein analysis, sequencing, and proteome characterization. In a typical experiment in bottom-up proteomics, proteins are extracted and proteolytically digested into peptides and separated by liquid chromatography. Peptide spectra are then collected using tandem mass spectrometry, within a cycle time of about 1 s[1]. Using this method, most of the proteins that have been expressed in an organism can be identified and quantified. However, proteins in biological samples are extremely heterogeneous, spanning several orders of magnitude in abundance. In addition, most eukaryote proteins contain a variegated and dynamic range of post-translational modifications (PTMs). Due to the fast amount of conceivable combinations, the identification and sequencing of proteins in such heterogeneous mixtures is challenging for conventional mass spectrometry[2].

A high-throughput single-molecule technique could address these limitations. Although no single-molecule protein sequencer exists today, a few approaches have been proposed, mainly aimed at protein identification. For instance, it has been shown that if only cysteine and lysine residues are read in sequence, most of human proteins can be identified[3]. In a recent proof-of-concept experiment[4], peptides with cysteine and a lysine residues were labeled with a fluorescence acceptor, whereas a ClpXP unfoldase/protease was labeled with a fluorescence donor. Then, single-molecule Förster resonance energy transfer was used to monitor the passage of the acceptor dyes near the donor dye as the linearized polypeptide was processively transported through the ClpXP chamber. In another recent method, millions of peptides with fluorescently labeled cysteine[5–7], lysine, or phosphoserine residues were immobilized on a glass coverslip. Total internal reflection fluorescence microscopy was then used to monitor each molecule's fluorescence following consecutive cycles of N-terminal amino acid removal using Edman degradation chemistry. The authors identified a variety of peptides and achieved single-molecule positional readout of the phosphorylated sites.

Nanopores might also be used for single-molecule protein analysis and sequencing. Stein and co-workers proposed to couple a nanopore to a mass spectrometer. The nanopore would linearize individual proteins, whereas the mass spectrometer would be used to identify peptides as they are sequentially cleaved[8]. In a more conventional nanopore approach, an external potential is applied across the nanopore and the resulting ionic current is used to recognize proteins or peptides traversing the nanopore. In an early experiment, inspired by DNA nanopore sequencing, a ClpXP enzyme complex was used to force the unfolding of a protein through a biological nanopore[9]. An independent study showed that nanopore currents are capable of recognizing modifications in individual amino acid within a linearized polypeptide strand[10]. However, despite these encouraging results enzymes that process proteins or polypeptides amino-acid-by-amino-acid are yet to be discovered.

In an alternative approach, a protease is placed atop of a nanopore to fragment a protein. Then the mass of individual peptides is identified by nanopore currents. This method would be similar to conventional protein sequencing using tandem mass spectrometry, with the additional advantage of being low-cost, portable, and single molecule. For this approach to be feasible, however, the signal rising from the peptide blockade must be directly correlated to the mass of the peptide. Previous work with PEG molecules[11–17], oligosaccharides[18], and homopolymeric peptides[19–21] revealed that there might be a direct correlation between the depth of the current blockade and the molecular weight of polymers, providing the charge composition of the analyte is uniform[22]. In such circumstances, it has been shown that nanopores can resolve the signal of poly-arginine peptides from 10 to 5 amino acids, hence distinguishing peptides differing by one arginine in length (174 Da)[19]. Peptides in a biological sample, however, have a heterogenous chemical composition. Work with DNA[23,24] and amino-acid enantiomers[25] revealed that the chemical identity of molecules and the charge inside the nanopore[26] have an unpredictable effect on the ionic current. On the other hand, additional work with peptides showed that the correlation between mass and ionic signal is retained with peptides[27,28], providing they are either neutral or uniformly charged. Nonetheless, peptides with an overall charge that is opposite to the applied bias have not been systematically studied, most likely because they are not efficiently captured and analyzed at such potentials[29–32]. Finally, the diameter and geometry of biological nanopores cannot be easily adapted to study the array of sizes, shapes, and chemical composition of polypeptides in solution.

Recently, we have shown that octameric fragaceatoxin C (FraC, Fig. 1a) nanopores[33] from the sea anemone *Actinia fragacea* can be used to study DNA[34], proteins, and peptides[35]. The trans-membrane region of FraC is unique compared with other nanopores used in biopolymer analysis as it is formed by α-helices that describe a sharp and narrow constriction at the *trans* exit of the nanopore. We showed that an electroosmotic flow across the nanopore can be engineered to capture polypeptides at a fixed potential despite their charge composition[35]. However, peptides smaller than 1.6 kDa in size translocated too fast across the nanopore to be sampled, indicating that nanopores with a smaller diameter should be used to detect peptides with lower molecular weight. In this work, we show that the diameter of FraC nanopores can be tuned, permitting the identification of a large range of peptides sizes. Using engineered nanopores, we also show that peptides differing by the substitution of one-amino acid (44 Da) can be identified. At selected pH conditions, the nanopore signal directly correlates to the mass of the peptide, including peptides with high content of acidic residues (i.e., negatively charged peptides at physiological pH). Therefore, this nanopore approach can be used to identify the mass of individual peptides in solution and, providing a protease is attached immediately above the nanopore, might allow the sequencing of proteins in real-time.

## Results

**Engineering the size of FraC nanopores.** One of the main challenges in biological nanopores analysis is to obtain nanopores with different size and shape. Most biological nanopores are formed by multiple repeats of individual monomers. Hence, different nanopore sizes might be obtained by engineering the protein oligomeric composition[36]. We noticed that at pH 7.5 a small fraction of wild-type FraC (Wt-FraC) nanopores showed a lower conductance (1.26 ± 0.08 nS, −50 mV, type II Wt-FraC) compared with the dominant fraction (2.26 ± 0.08 nS, −50 mV, type I Wt-FraC), suggesting that FraC might be able to spontaneously assemble into nanopores with a smaller size. At pH 4.5, type I and type II FraC nanopores were also observed, however, a smaller nanopore conductance was identified alongside (0.42 ± 0.03 nS, type III Wt-FraC, −50 mV, Fig. 1b). Occasionally, nanopores with a yet smaller conductance were observed, however, their appearance was too rare for meaningful characterization. We noticed that the reconstitution of lower conductance nanopores depended on several purification conditions (Supplementary Figure 1 and 2). In particular, the occurrence of type II and type III nanopores increased when the oligomers were stored in solution for several weeks or when the concentration of monomeric Wt-FraC was reduced during oligomerization (Supplementary Figure 1 and 2). In an effort to enrich type II and type

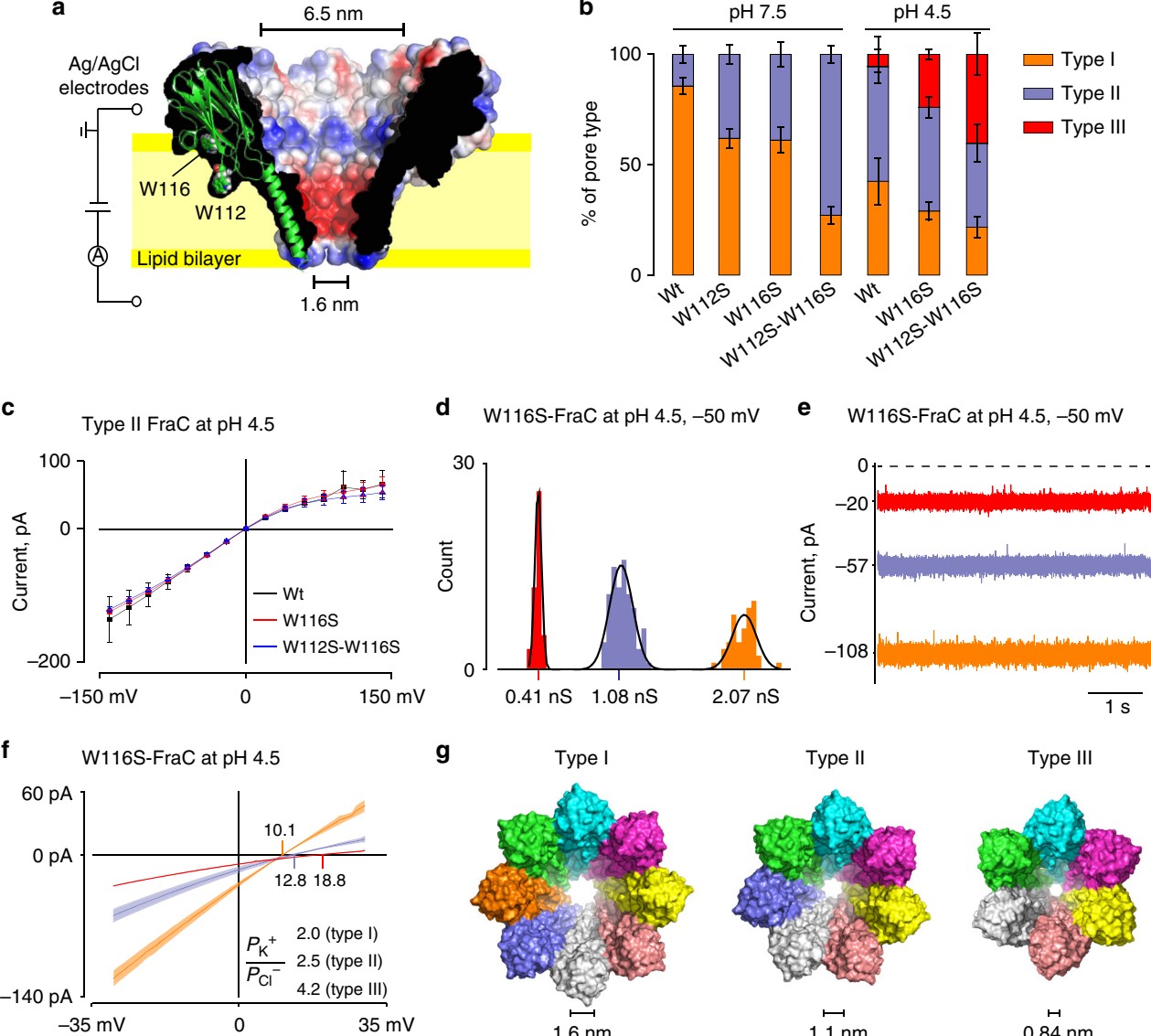

**Fig. 1** Preparation and characterization of type I, type II, and type III fragaceatoxin C (FraC) nanopores. **a** Cut through of a surface representation of wild-type FraC (Wt-FraC) oligomer (PDB: 4TSY[33]) colored according to the vacuum electrostatic potential as calculated by PyMOL. One protomer is shown as a carton presentation with tryptophans 112 and 116 displayed as spheres. **b** Percentage of the distribution of type I, type II, and type III for Wt-FraC, W112S-FraC, W116S-FraC, and W112S-W116S-FraC at pH 7.5 and 4.5. **c** IV curves of type II nanopores formed by Wt-FraC, W116S-FraC, and W112S-W116S-FraC at pH 4.5. **d** Single nanopore conductance of W116S-FraC in 1 M KCl at pH 4.5 and −50 mV. **e** Typical current traces for the three nanopore types of W116S-FraC in 1 M KCl at pH 4.5 under −50 mV applied potential. **f** Reversal potentials measured under asymmetric condition of KCl (1960 mM *cis*, 467 mM *trans*) at pH 4.5 for the three W116S-FraC nanopore types. The ion selectivity was calculated using the Goldman–Hodgkin–Katz equation (Eq. 1)[52]. **g** Molecular models of the three type FraC nanopores constructed from the FraC crystals structure using the symmetrical docking function of Rosetta. The diameters were measured from the distance between opposite side chains of D10 and include the van der Waals radii of the atoms. The electrophysiology recordings were performed with a 10 kHz sampling and a 2 kHz Bessel filter. The error bars and color shadow in the I–V curves are standard deviations from at least three repeats

III FraC nanopores, we weakened the interaction between the nanopore and the lipid interface by substituting W112 and W116 at the lipid interface of FraC (Fig. 1a) with serine. We reasoned that a lower concentration of monomers, during oligomerization, would increase the population of lower molecular mass oligomers. Rewardingly, we found that at both pH 7.5 and pH 4.5 the proportion of type II and type III FraC nanopores increased dramatically. For example, W112S-W116S-FraC formed 60% of type II pore at pH 7.5, and 40% of type III pore at pH 4.5 (Fig. 1b, Supplementary Figure 3). The different nanopore types could also be separated by Ni-NTA affinity chromatography using an imidazole gradient (Supplementary Figure 2e-f). Finally, at pH 7.5,

type II and type III FraC nanopores could also be obtained by replacing aspartic acid 109 (Supplementary Note 1, Supplementary Figure 2g-h, 3e-f) at the lipid interface with serine. Importantly, the reconstituted type II and type III nanopores did not show any particular gating (spontaneous opening and closing) or bilayer instability (e.g., the detachment of the nanopores from the lipid bilayer was never observed).

Among FraC nanopores of the same type, the lipid interface modifications brought by W112S and W116S substitutions did not alter the conductance and ion selectivity of the nanopores (Fig. 1c, Supplementary Figure 3 and 4, Supplementary Table 1), suggesting that the overall fold of the nanopores was unchanged

by the surface modifications. When characterized in lipid bilayers, type I, type II, and type III nanopores showed a well-defined single conductance distribution, a steady open pore current (Figs. 1d, e) and comparable power spectra (Supplementary Figure 5). Interestingly, the nanopore types with a reduced conductance also showed an increased cation selectivity (2.0±0.1, 2.5±0.2, and 4.2±0.2 for type I, type II, type III W116S-FraC nanopores, respectively, at pH 4.5, Fig. 1f, Supplementary Table 1). The increased ion selectivity most likely reflects a larger overlap of the electrical double layer in the nanopores with a narrower constriction. These and several addition lines of evidence (Supplementary note 1, Supplementary Figure 6) strongly suggest that the three types of FraC nanopores represent nanopores with different protomeric compositions. Molecular modeling allowed predicting the diameter of type II (1.1 nm) and type III (0.84 nm) nanopores (Fig. 1g). These values corresponded well to the diameters estimated from their conductivity values (1.17 ± 0.04 and 0.71 ± 0.01 for type II and type III, Supplementary Figure 3). Notably, type III FraC, having a sub-nanometer constriction, is the biological nanopore with the smallest inner diameter known to date.

### Identification of single amino-acid substitutions with type II FraC nanopores.

Type II FraC nanopores were used to sample a series of angiotensin peptides (Figs. 2–3, Table 1, Supplementary Figure 7), which regulate blood pressure and fluid balance. The peptides were added to the *cis* side of type II W116S-FraC nanopores and the magnitude of the ionic current associated with a peptide blockade ($I_B$) was measured. The pH of the solution was set to 4.5, because at higher pH the capture of some peptides was either not observed or greatly reduced[35]. To characterize the

peptide blockade, we used the percentage of excluded currents ($I_{ex\%}$), defined as $[(I_O - I_B)/I_O] \times 100$, where $I_O$ represents the open pore current. $I_{ex\%}$, which relates to the ionic current that is lost during the transit of the peptide across the nanopore, and is expected to be proportional to the volume inside the nanopore excluded by the peptide. Angiotensin I (DRVYIHPFHL, 1296.5 Da), showed the deepest blockade ($I_{ex\%} = 91.2 \pm 0.2$) and angiotensin IV (VYIHPF, 774.9 Da) the shallowest blockade ($I_{ex\%} = 61.1 \pm 4.0$). The percent of excluded current of angiotensin II (DRVYIHPF, 1046.2 Da, $I_{ex\%} = 82.1 \pm 1.3$) and angiotensin III (RVYIHPF, 931.1 Da, $I_{ex\%} = 77.9 \pm 0.5$) fell at intermediate values. When the four peptides were tested simultaneously, individual peptides could be discriminated (Fig. 2f).

The resolution limit of the nanopore sensor was challenged by sampling mixtures of angiotensin II and angiotensin A, which have an identical composition with the exception of the initial amino acid that is aspartate in angiotensin II and alanine in angiotensin A. These two peptides, differing by 44 Da, appeared as distinctive peaks in $I_{ex\%}$ plots (Fig. 3). Smaller peptide differences, e.g., the 34 Da difference between phenylalanine and isoleucine in angiotensin III and Ile7 angiotensin III, were observed but not easily detected (Supplementary Figure 8), placing the resolution of our system at ~ 40 Da. It should be noticed that a more complex classification of peptides has been demonstrated elsewhere[37–39], and would likely improve the sensitivity of discrimination. Smaller peptides such as angiotensin II 4–8 (YIHPF, 675.8 Da), endomorphin I (YPWF, 610.7 Da), or leucine enkephalin (YGGFL, 555.6 Da) translocated too quickly across type II W116S-FraC nanopores to be sampled, but they could be measured using type III W112S-W116S-FraC nanopores (Table 1, Supplementary Note 2, Supplementary Figure 9-11).

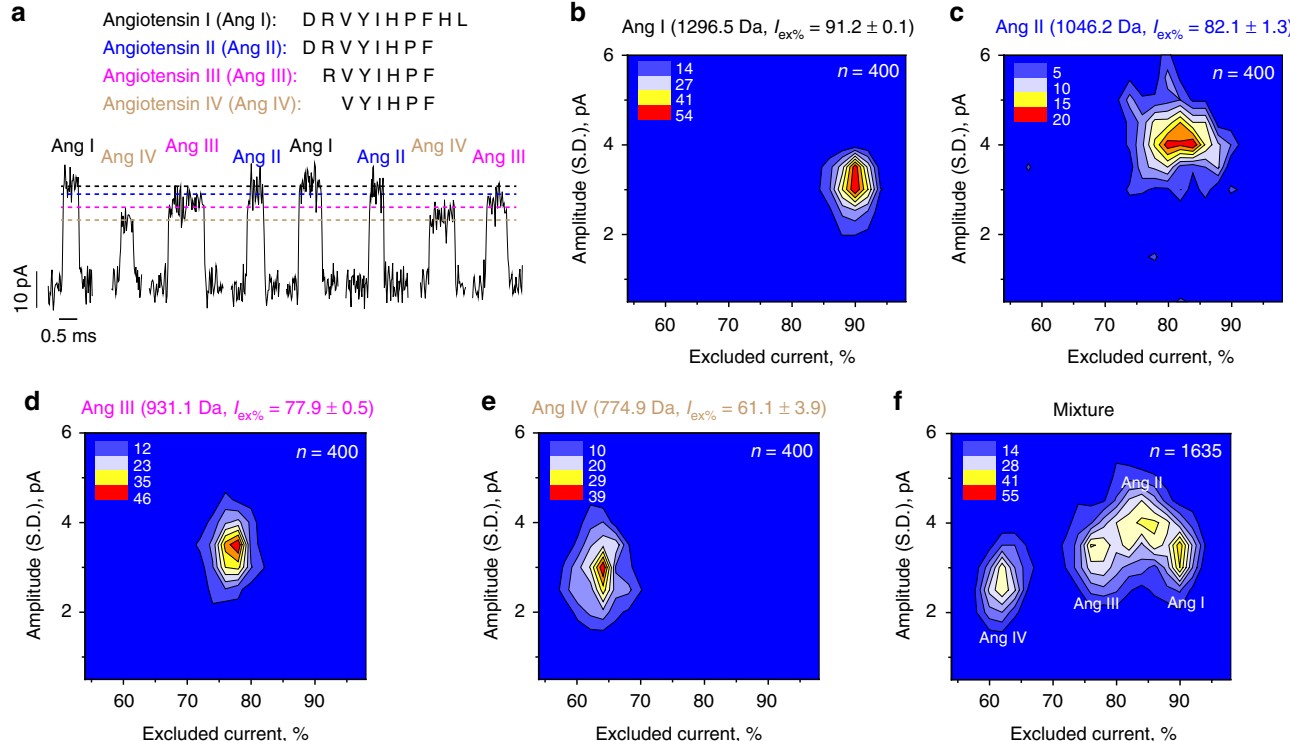

**Fig. 2** Discrimination of angiotensin peptides using type II W116S-fragaceatoxin C (FraC) nanopores at pH 4.5. **a** Peptide sequences of angiotensin I (Ang I), angiotensin II (Ang II), angiotensin III (Ang III), and angiotensin IV (Ang IV) and typical blockades provoked by the four angiotensin peptides measured at −30 mV. **b–e** Color density plot of the $I_{ex\%}$ versus the standard deviation of the current amplitude for angiotensin I, II, III, and IV, respectively. **f** Discrimination of four angiotensin peptides in a mixture. Peptides were added into the *cis* chamber and measured at −30 mV. Standard deviations were calculated from at least three independent repeats. Color density plots were created using Origin

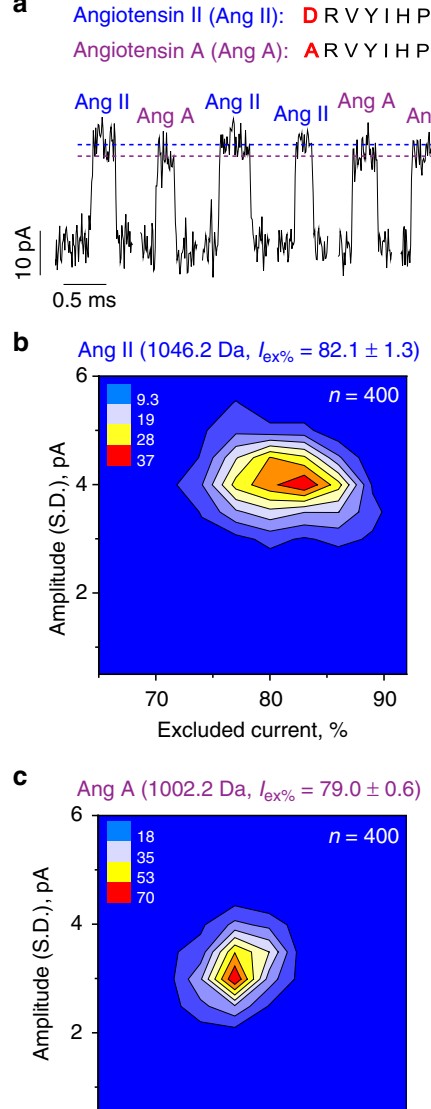

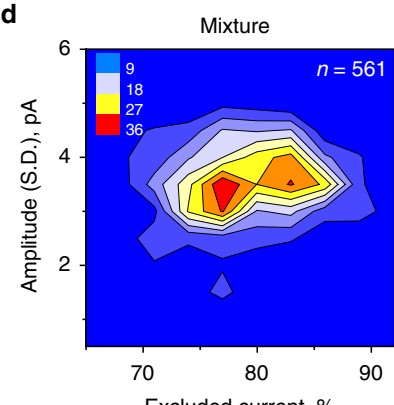

**Fig. 3** Discrimination of peptides differing by a single amino acid using type II W116S-fragaceatoxin C (FraC) at pH 4.5. **a** Peptide sequences of angiotensin II, and A with typical blockades provoked by the two angiotensin peptides measured at −30 mV applied bias. **b**, **c** Color density plot of the $I_{ex\%}$ versus the standard deviation of the current amplitude for angiotensin II, and A, respectively. **d** Separation of angiotensin II and A in a mixture. Peptides were added into the *cis* chamber and measured under −30 mV. Standard deviations were calculated from at least three independent repeats

holding prior knowledge of the analyte identity. In nanopores, ionic current blockades are expected to be directly proportional to the volume excluded by the analyte inside the nanopore[40]. Hence, the current blockade of a peptides should reveal the volume of the peptide, which might approximate to its mass by the relation: volume $(nm^3) = 1.212 \times 10^{-3}$ $(nm^3/Da) \times MW$ (Da)[41,42]. In the effort to assess FraC nanopores as a peptide mass identifier, we tested additional peptides at pH 4.5 in 1 M KCl solutions using type I, type II, and type III FraC nanopores (Figs. 4a–c, Table 1, Supplementary Figure 7, 10, 12). We found that for most peptides there was a direct correlation between the excluded current and the volume/mass of the peptide. Although linear regression fitted the data well, if the expected values for an empty nanopore were to be included (i.e., $I_{ex\%}$ is zero when no peptide is inside the nanopore), quadratic functions showed best fits for the data collected with type I and type II nanopores (Figs. a, b). By contrast, linear regressions could be used for the data measured with type III FraC nanopores (Fig. 4c). Interestingly, the extrapolated volumes for a fully occupied nanopore (3.5 $nm^3$, 2.0 $nm^3$, and 0.96 $nm^3$ for type I, type II, and type III FraC, respectively), were similar to the volumes comprised between D10 and D17 residues of FraC (3.6 $nm^3$, 1.8 $nm^3$, and 1.0 $nm^3$, respectively, Fig. 4), suggesting that the constriction (D10) and the amino acid one turn of the helix above it (D17) most likely define the sensing region within the nanopore.

Although the $I_{ex\%}$ of most peptides fitted well to the empirical quadratic functions, two notable exceptions were c-Myc 410–419 (1203.3 Da) and neuropeptide-like protein 3 (NLP-3) (66–75, 1099.2 Da). These peptides were intentionally selected because they included several acidic residues (Table 1). c-Myc 410-419 and NPL-3 (added in *cis*) could be readily captured at negative applied potentials (*trans*), indicating that the *cis* to *trans* electroosmotic flow across the nanopore can overcome the electrostatic energy barrier opposing peptide capture. However, the dwell times were faster and the $I_{ex\%}$ lower than peptides with similar mass (Table 1, Fig. 4b), suggesting that electrophoretic and electrostatic interactions between the pore and the peptides might prevent them from entering the sensing region of the nanopore.

Thus, we tested a range of pHs where the aspartate and glutamate side chains are expected to be protonated (Fig. 5a, Table 1). We found that only at pH 3.8, the signal corresponding to c-Myc 410–419 (1203.3 Da) fell between the signal of angiotensin I (1296.5 Da), and angiotensin II (1046.2 Da, Fig. 5a), suggesting that after losing its negative charges, c-Myc 410–419 peptide might access the recognition volume of FraC. Rewardingly, at pH 3.8 all the remaining peptides showed $I_{ex\%}$ values that scaled with the masses of the peptides (Fig. 5b). Notably, at pH 3.8 the peptide signals showed relatively high variability and the conditions had to be carefully controlled (Supplementary Note 3). Most likely, this is because at pH 3.8 the charge density of the constriction (Fig. 1a) is strongly affected by small variations in pH.

**A nanopore mass spectrometer for peptides**. Although the ability of nanopores to distinguish between known analytes is useful, a more powerful application would be the identification of peptide masses directly from ionic current blockades without

**Table. 1 Peptide analysis with different types of FraC nanopores at pH 4.5**

| Peptide | Sequence | Molecular weight (g/mol) | Volume (nm³) | Charge | | $I_{ex\%}$ pH 4.5 | Dwell time (ms) |
|---|---|---|---|---|---|---|---|
| | | | | pH 4.5 | pH 3.8 | | |
| Wt-FraC type I pore, −30 mV | | | | | | | |
| Endothelin 2 | CSCSSWLDKECVYFCHLDIIW | 2546.9 | 3.087 | 0.36 | 1.56 | 93.9 ± 1.8 | 104.0 ± 29.9 |
| Endothelin 1 | CSCSSLMDKECVYFCHLDIIW | 2491.9 | 3.020 | 0.36 | 1.56 | 92.5 ± 0.5 | 19.73 ± 1.95 |
| Dynorphin A | YGGFLRRIRPKLKWDNQ | 2147.5 | 2.599 | 4.48 | 4.97 | 84.9 ± 2.6 | 3.68 ± 0.76 |
| Pre-angiotensinogen 1–14 | DRVYIHPFHLVIHN | 1758.9 | 2.130 | 3.45 | 3.96 | 75.4 ± 2.3 | 0.29 ± 0.04 |
| Angiotensin I | DRVYIHPFHL | 1296.5 | 1.568 | 2.46 | 2.96 | 56.6 ± 0.9 | 0.15 ± 0.04 |
| W116S-FraC type II pore, −30 mV | | | | | | | |
| Angiotensin I | DRVYIHPFHL | 1296.5 | 1.568 | 2.46 | 2.96 | 91.2 ± 0.2 | 0.54 ± 0.01 |
| c-Myc 410–419 | EQKLISEEDL | 1203.3 | 1.456 | −1.19 | 0.36 | 70.0 ± 3.4 | 0.12 ± 0.01 |
| Angiotensin 1–9 | DRVYIHPFH | 1183.3 | 1.431 | 2.46 | 2.96 | 86.0 ± 0.2 | 0.37 ± 0.04 |
| NLP-3 (66–75) | YFDSLAGQSL | 1099.2 | 1.331 | −0.52 | 0.97 | 75.3 ± 3.0 | 0.23 ± 0.02 |
| Angiotensin II | DRVYIHPF | 1046.2 | 1.266 | 1.47 | 1.96 | 82.1 ± 1.3 | 0.37 ± 0.04 |
| Asn1Val5 AngioII | NRVYVHPF | 1031.2 | 1.248 | 2.03 | 2.16 | 80.4 ± 0.2 | 0.34 ± 0.06 |
| Angiotensin A | ARVYIHPF | 1002.2 | 1.212 | 2.03 | 2.16 | 79.0 ± 0.6 | 0.34 ± 0.02 |
| Angiotensin III | RVYIHPF | 931.1 | 1.127 | 2.03 | 2.16 | 77.9 ± 0.5 | 0.35 ± 0.04 |
| Ile7 Angiotensin III | RVYIHPI | 897.1 | 1.085 | 2.03 | 2.16 | 75.7 ± 0.4 | 0.19 ± 0.05 |
| Angiotensin IV | VYIHPF | 774.9 | 0.938 | 1.02 | 1.16 | 61.1 ± 4.0 | 0.15 ± 0.06 |
| W112S-W116S-FraC type III pore, −50 mV | | | | | | | |
| Angiotensin IV | VYIHPF | 774.9 | 0.938 | 1.02 | 1.16 | 98.9 ± 0.8 | 0.61 ± 0.07 |
| Angiotensin 4–8 | YIHPF | 675.8 | 0.818 | 1.02 | 1.16 | 91.8 ± 0.4 | 0.40 ± 0.04 |
| Endomorphin I | YPWF | 610.7 | 0.741 | 0.04 | 0.17 | 80.3 ± 0.5 | 0.32 ± 0.04 |
| Met5 Enkephalin | YGGFM | 573.7 | 0.695 | 0.04 | 0.17 | 66.5 ± 0.7 | 0.16 ± 0.02 |
| Leucine Enkephalin | YGGFL | 555.6 | 0.673 | 0.04 | 0.17 | 65.6 ± 2.4 | 0.20 ± 0.05 |

The charges of the peptides were calculated according to the pKa for individual amino acid[53]. Standard deviations were obtained for at least three measurements
FraC fragaceatoxin C, Wt-FraC wild-type FraC

**Peptide translocation across nanopores**. It has been assumed[43–46] and experimentally[47] proven that the voltage dependence of the average dwell time ($\tau_{off}$) can report on the translocation of a molecule across a nanopore. Under a negative bias (*trans*) for positively charged peptides (added in *cis*), both electrophoretic and electroosmotic forces (from *cis* to *trans*) promote the entry and translocation[35] across the nanopore (Supplementary Figure 13). For negatively charged peptides, such as c-Myc 410–419 at pH 4.5 (Fig. 5a), the electroosmotic driving force must be stronger than the opposing electrophoretic force. The voltage dependence of $\tau_{off}$ was then examined for the most acidic peptide c-Myc 410–419 at different pH values (Fig. 5c). At pH 4.5, the peptide exhibited a maximum in $\tau_{off}$ at −50 mV, suggesting that at low potentials c-Myc 410–419 returns to the *cis* chamber (<50 mV), and at higher potentials (>50 mV) c-Myc 410–419 exits to the *trans* chamber. At pH 3.8 and lower, we observed a decrease in $\tau_{off}$, albeit at higher potentials, indicating that at pH 3.8 c-Myc 410–419 crossed the membrane region of FraC to the *trans* chamber.

## Discussion

We have engineered the assembly of FraC to obtain three nanopores types with 1.6, 1.1, and 0.84 nm inner diameters. The nanopores can accommodate peptides ranging from 22 to 4 amino acids in length. Smaller peptides might be detected using further fine tuning of the transmembrane region of the nanopore, for example, by introducing amino acids with bulky side chains in the recognition volume of the nanopore. We also showed that the nanopores can discriminate differences between an alanine and a glutamate (44 Da) in a mixture of peptides. Furthermore, we found that at exactly pH 3.8 the ionic signal of the peptides depended on the mass of the analyte, whereas at higher pH values the current signal of negatively charged peptides was higher than expected from their mass alone. Most likely, a negatively charged recognition region is important for creating an electrostatic environment for peptide mass recognition. At the same time, the electrostatic interaction of the constriction with negatively charged analytes might prevent the correct positioning of the analyte within the reading frame of the nanopore. Hence, the next-generation nanopores might be fabricated using unnatural amino acids that hold a negative charge at a low pH range (e.g., sulfate or phosphate groups). Alternatively, peptides might be chemically modified (e.g., by esterification) to neutralize the negative charge.

Mass spectrometry is the workhorse of the proteomics field. At present, the nanopore system falls short from the resolution of commercial mass spectrometers. However, the technology is young and improvements are to be expected. It should also be noticed that a peptide mass analyzer device based on nanopores will have distinctive advantages compared with conventional mass spectrometers, the latter being expensive, extremely complex, and unwieldy. By contrast, nanopores can be integrated in portable and low-cost devices containing hundreds of thousands of individual sensors. In addition, the electrical nature of the signal allows sampling biological samples in real-time. Furthermore, since the nanopore reads individual molecules, the signal contains additional information not available for ensemble techniques. Finally, single-molecule detection, especially when coupled to high-throughput analysis, is amenable for detecting low abundance peptides and to unravel the chemical heterogeneity in PTMs, challenges that are hard to address with conventional mass spectrometry.

A nanopore peptide mass detector might also be integrated in real-time protein sequencing system, providing a number of requirements are met. First, a protease-unfoldase pair should be coupled directly above the nanopore sensor. The barrel-shaped ATP-dependent ClpXP protease appears to be an ideal candidate because it would encase the digested peptides preventing its

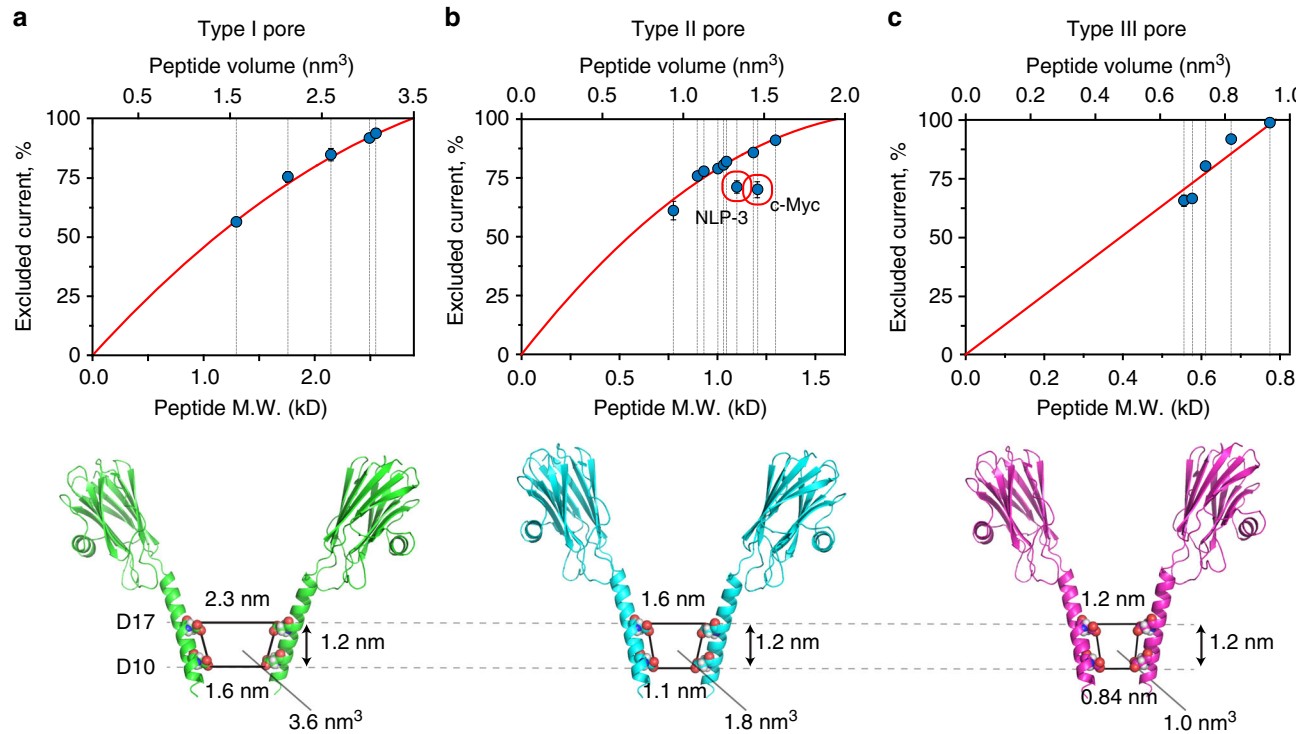

**Fig. 4** Recognition of peptides with different chemical composition at pH 4.5. On the top graph is the relation between the molecular weight (M.W.) or volume of the peptide and the $I_{ex\%}$. The bottom figure shows the sensing volume of type I wild-type fragaceatoxin C (Wt-FraC) (**a**), type II W116S-FraC (**b**), and type III W112S-W116S-FraC (**c**) nanopores. The solid line represents a second order polynomial fitting in **a**, **b** and a linear fitting in **c**, with the extrapolated value at 100% $I_{ex\%}$ corresponding to the volume of a peptide that would completely occupy the sensing volume of the nanopore. The latter is most likely constricted to the volume included between the constriction of the pore (aspartic acid 10) and the residues that lie one turn of a helix above the constriction (aspartic acid 17). The distances are measured from two opposing residues and include the van der Waals radii of the atoms. Current blockades were measured at −30 mV for type I and II pore, and at −50 mV for type III pore in 1 M KCl solutions. The error bars represent standard deviation from at least three repeats. Red circles highlight the two peptides that bare a negative charge at pH 4.5 (Table 1)

release in solution. The coupling could be achieved by chemical attachment, by genetic fusion, or by introducing binding loops to the nanopore that interact with the peptidase. We have taken the latter approach to couple α-hemolysin nanopores with hepta-meric GroEL[48]. The cleaved peptides will be sequentially recognized and translocated across the nanopore. Here we have taken several steps showing this approach might be feasible. We demonstrated that the peptides entering the *cis* side of the nanopore have a high probability of exiting the nanopore to the *trans* chamber, which will prevent duplicate detection events. Furthermore, we showed that at pH 3.8 peptides are likely to be captured and their mass recognized by the nanopore at a fixed applied potential irrespectively of their chemical composition. If such low pH values will not be compatible with enzymatic activity, asymmetric solutions on both side of the nanopore can be used[49–51]. In such system, conditions in the *cis* side will be tuned to optimize the ATPase activity of the unfoldase-peptidase, whereas the pH and ionic strength of the *trans* side will be optimized to capture and recognize individual peptides.

## Methods

**Chemicals**. Endothelin 1 (≥97%, CAS# 117399-94-7), endothelin 2 (≥97%, CAS# 123562-20-9), dynorphin A porcine (≥95%, CAS# 80448-90-4), angiotensin I (≥90%, CAS# 70937-97-2), angiotensin II (≥93%, CAS# 4474-91-3), c-Myc 410-419 (≥97%, # M2435), Asn1-Val5-Angiotensin II (≥97%, CAS# 20071-00-5), Ile7 Angiotensin III (≥95%, #A0911), leucine enkephalin (≥95%, #L9133), 5-methionine enkephalin (≥95%, CAS# 82362-17-2), endomorphin I (≥95%, CAS# 189388-22-5), pentane (≥99%, CAS# 109-66-0), hexadecane (99%, CAS# 544-76-3), Trizma®hydrochloride (≥99%, CAS# 1185-53-1), Trizma®base (≥99%, CAS# 77-86-1), potassium chloride (≥99%, CAS# 7447-40-7), N,N-

dimethyldodecylamine N-oxide (LADO, ≥99%, CAS# 1643-20-5) were obtained from Sigma-Aldrich. Pre-angiotensinogen 1–14 (≥97%, # 002-45), angiotensin 1–9 (≥95%, # 002-02), angiotensin A (≥95%, # 002-36), angiotensin III (≥95%, # 002-31), angiotensin IV (≥95%, # 002-28) NLP-3 (66–75) (≥97%, # 076-36) were purchased from Phoenix Pharmaceuticals. Angiotensin 4–8 (≥95%) was synthesized by BIOMATIK. 1,2-Diphytanoyl-sn-glycero-3-phosphocholine (DPhPC, #850356P) and sphingomyelin (Porcine brain, # 860062) were purchased from Avanti Polar Lipids. Citric acid (99.6%, CAS# 77-92-9) was obtained from ACROS. n-Dodecyl β-D-maltoside (DDM, ≥99.5%, CAS# 69227-93-6) was bought from Glycon Biochemical EmbH. DNA primers were synthesized from Integrated DNA Technologies (IDT), enzymes from Thermo Scientific. All peptides were dissolved with Milli-Q water without further purification and stored in −20 °C freezer. pH 7.5 buffer containing 15 mM Tris in this study was prepared by dissolving 1.902 g Trizma® HCl and 0.354 g Trizma® base in 1 liter Milli-Q water (Millipore, Inc.).

**FraC monomer expression and purification**. FraC gene containing NcoI and HindIII restriction sites at the 5′ and 3′ ends, respectively, and a sequence encoding for a poly-histidine tag at the 3′ terminus was cloned into a pT7-SC1 plasmid. Plasmids were transformed into BL21(DE3) E.cloni® competent cell by electro-poration. Cells were grown on lysogeny broth (LB) agar plate containing 100 μ/mL ampicillin overnight at 37 °C. The entire plate was then harvested and inoculated into 200 mL fresh 2YT media and the culture was grown with 220 rpm shaking at 37 °C until the optical density at 600 nm of the cell culture reached 0.8. Then, 0.5 mM isopropyl β-D-thiogalactoside (IPTG) was added to the media and the culture was transferred to 25 °C for overnight growth with 220 rpm shaking. The next day, the cells were centrifuged (2000×g, 30 min) and the pellet stored at −80 °C. FraC was purified from cell pellets harvested from 100 mL culture media. In all, 30 mL of cell lysis buffer (150 mM NaCl, 15 mM Tris, 1 mM MgCl₂, 4 M urea, 0.2 mg/mL lysozyme, and 0.05 unit/mL DNase) were added to re-suspend the pellet and vigorously mixed for 1 h. Cell lysate was then sonicated with Branson Sonifier 450 for 2 min (duty cycle 10%, output control 3). Afterwards, the crude lysate was centrifuged down at 4 °C for 30 min (5400 × g), and the supernatant incubated with 100 μL Ni-NTA beads (Qiagen) for 1 h with gentle shaking. Beads were spun down and loaded to a Micro Bio-spin column (Bio-Rad). In total, 10 mL

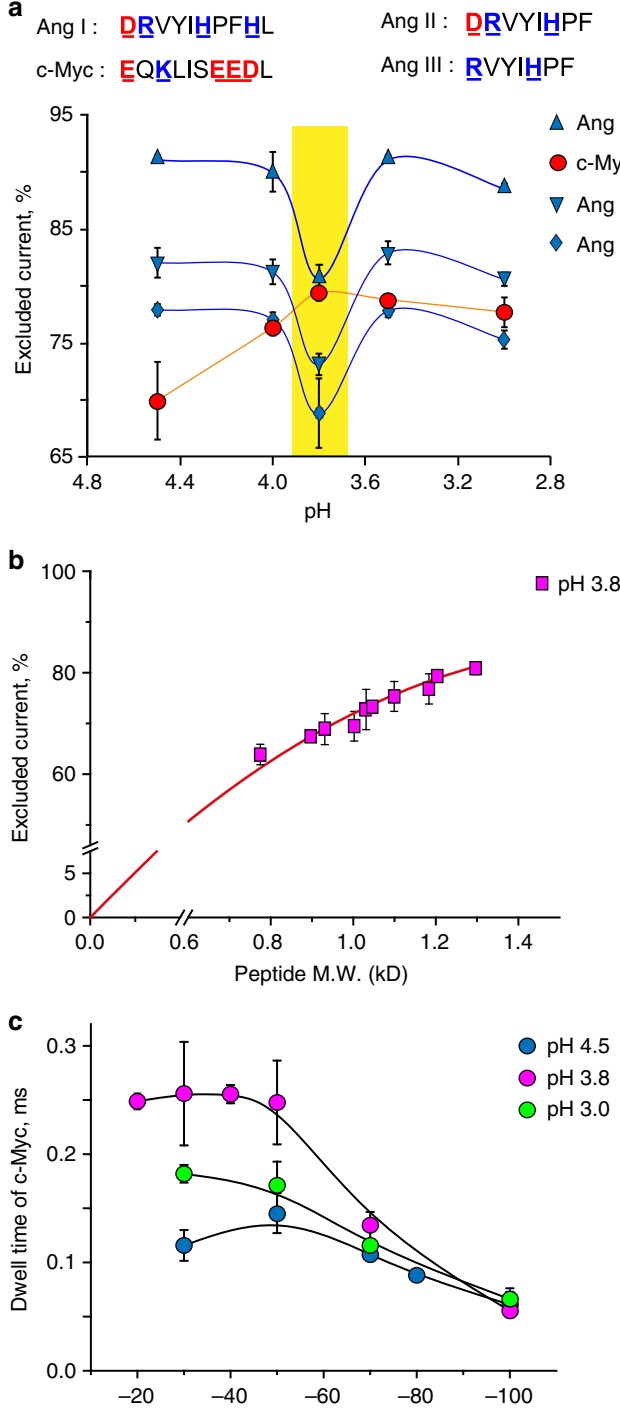

**Fig. 5** A nanopore peptide mass identifier. **a** Top, sequence of the four peptides tested. The amino acids that have a positive charge are in blue and the acidic residues in red. Below, pH dependence of the $I_{ex\%}$ for the four peptides (*cis*) using type II W116S-fragaceatoxin C (FraC) nanopores under −30 mV applied potential. **b** Relationship between the $I_{ex\%}$ and the mass of peptides at pH 3.8. **c** Voltage dependence of c-Myc dwell times at different pHs. All electrophysiology measurements were carried out in 1 M KCl, 0.1 M citric acid. The charges of the peptides were calculated according to the $pK_a$ for individual amino acids[53]. Standard deviations were calculated from at least three independent repeats

of SDEX buffer (150 mM NaCl, 15 mM Tris, pH 7.5) containing 20 mM imidazole was used to wash the beads, and proteins were eluded with 150 μL elution buffer (SDEX buffer, 300 mM imidazole). The concentration of the protein was determined by the absorption at 280 nm with Nano-drop 2000 (Thermo Scientific) using the elution buffer as blank. To further confirm the purity of monomer, the protein solution was diluted to 0.5 mg/mL using the elution buffer and 9 μL of the diluted sample was mixted with 3 μL of 4× loading buffer (250 mM Tris HCl, pH 6.8. 8% SDS, 0.01% bromophenol blue and 40% glycerol) and then loaded to 12% sodium dodecyl sulfate-polyacrylamide gel electrophoresis (SDS-PAGE). Gels were run under a constant applied current of 35 mA for 30 min, and stained with Coomassie dye (InstantBlue™, Expdedeon) before viewing using a gel imager (Gel Doc™, Bio-Rad).

**FraC mutation preparation.** FraC mutants were prepared according to MEGA-WHOP method[54]. In all, 25 μL REDTaq® ReadyMix™ was mixed with 4 μM primer (Supplementary Figure 2) containing the desired mutation with 50 ng plasmid (pT7-SC1 with Wt-FraC gene) as template and the final volume was brought to 50 μL with Milli-Q water. The PCR protocol was initiated by a 150-s denature step at 95 °C, followed by 30 cycles of denaturing (95 °C, 15 s), annealing (55 °C, 15 s), and extension (72 °C, 60 s). The PCR products (MEGA primer) were combined and purified using a QIAquick PCR purification kit with a final DNA concentration around 200 ng/μL. Then, a second PCR was performed using the MEGA primer for whole-plasmid amplification. In all, 2 μL of MEGA primer, 1 μL Phire II enzyme, 10 μL 5× Phire buffer, 1 μL 10 mM dNTPs, were mixed with PCR water to 50 μL final volume. PCR started with pre-incubated at 98 °C (30 s) and then 25 cycles of denaturing (98 °C, 5 s), extension (72 °C, 180 s). When the PCR was completed, 1 μL DpnI enzyme was added and the mixture kept at 37 °C for 1 h. Then, the temperature was raised to 65 °C for 1 min to inactivate the enzyme. Products were then transformed into E. cloni® 10G (Lucigen) competent cell by electroporation. Cells were grown on LB agar plates containing 100 μg/mL ampicillin and were grown overnight at 37 °C. Single clones were enriched and sent for sequencing.

**Sphingomyelin-DPhPC liposome preparation.** In all, 20 mg sphingomyelin and 20 mg DPhPC (1,2-diphytanoyl-sn-glycero-3-phosphocholine) were dissolved in 4 mL pentane with 0.5% v/v ethanol and brought to a round flask. The solvent was then removed by rotation while being heated using a hair dryer. After evaporation, the flask was kept at ambient temperature for an additional 30 min. The lipid film was resuspended with 4 mL SDEX buffer (150 mM NaCl, 15 mM Tris, pH 7.5) and the solution immersed in a sonication bath for 5 min. Liposome suspensions were stored at −20 °C.

**FraC oligomerization.** FraC oligomerization was triggered by incubation of FraC monomers with sphingomyelin-DPhPC liposomes. Frozen liposome were thawed and sonicated in a water bath for 1 min. FraC monomers were diluted to 1 mg/mL using SDEX buffer, and then 50 μL of FraC monomers were added to 50 μl of a 10 mg/mL liposome solution to obtain a mass ratio of 10:1 (liposome:protein). The lipoprotein solution was incubated at 37 °C for 30 min to allow oligomerization. Then, 10 μl of 5% (w/v, 0.5% final) N,N-Dimethyldodecylamine N-oxide (LDAO) was added to the lipoprotein solution to solubilize the liposomes. After clarification (typically 1 min), the solution was transferred to a 50 mL Falcon tube. Then, 10 mL of SDEX buffer containing 0.02% DDM and 100 μL of pre-washed Ni-NTA beads were added to the Falcon tube and mixed gently in a shaker for 1 h at room temperature. The beads were then spun down and loaded to a Micro Bio-spin column. In all, 10 mL wash buffer (150 mM NaCl, 15 mM Tris, 20 mM imidazole, 0.02% DDM, pH 7.5) was used to wash the beads and oligomers eluded with 100 μL elution buffer (typically 200 mM EDTA, 75 mM NaCl, 7.5 mM Tris pH 7.5, 0.02% DDM). The FraC oligomers were stored at 4 °C and the nanopores are stable for several months.

**W112S-W116S-FraC oligomer separation with His-Trap chromatography.** In total, 200 μL of W112S-W116S-FraC monomers (3 mg/mL) were incubated with 300 μL of Sphingomyelin-DPhPC liposome (10 mg/mL) and kept at 4 °C for 48 h after which 0.5% LADO (final concentration) was added to solubilize the lipoprotein. Then the buffer was exchanged to 500 mM NaCl, 15 mM Tris, 0.01% DDM, 30 mM imidazole, pH 7.5 (binding buffer) using a PD SpinTrap G-25 column. W112S-W116S-FraC oligomers were then loaded to Histrap HP 1 mL column (General Electric) using an ÄKTA pure FPLC system (General Electric). The loaded oligomers were washed with 10 column volumes of 500 mM NaCl, 15 mM Tris, 0.01% DDM, 30 mM imidazole, pH 7.5, prior to applying an imidazole gradient (from 30 mM to 1 M imidazole, 500 mM NaCl, 15 mM Tris, 0.01% DDM, pH 7.5) over 30 column volumes. The protein concentration in flow was monitored with the absorbance at 280 nm and fractions were collected when the absorbance was higher than 5 mAu.

**Electrophysiology measurement and data analysis.** Electrical recordings were performed using two silver/silver-chloride electrodes immersed into an electrophysiology chamber connected to an Axopatch 200B amplifier (Axon Instrument). The chamber was separated into two 500 μL compartments by a ~ 100 μm

polytetrafluoroethylene Teflon aperture (Goodfellow Cambridge Limited). The aperture was pretreated with ~ 5 μL of hexadecane (10% v/v hexadecane in pentane) before loading the buffer. A bilayer was formed using 10 μL of 10 mg/mL DPhPC solution (in pentane), which was added into each compartment[35,55]. Ionic currents were digitized with a Digidata 1440 A/D converter (Axon Instrument). All peptides measurements were conducted with a 50 kHz sampling rate and a 10 kHz Bessel filter. Single-channel events were collected by applying the single-channel search function in Clampfit (Molecular Devices). Events shorter than 100 μs were ignored. $I_O$ values, referring to open pore current, were measured by using Gaussian fittings to event amplitude histograms. Percent of excluded current values ($I_{ex\%}$) were calculated by dividing the excluded current ($I_O - I_B$) by open pore current ($I_O$) and multiplied by 100. Dwell times and interevent times were measured by fitting single exponentials to histograms of cumulative distribution. Electrical recordings at pH 7.5 were performed using 1 M NaCl solutions and 15 mM Tris, recordings at pH 4.5 were performed using 1 M KCl solutions in 0.1 M citric acid and 180 mM Tris base.

**Ion permeability measurement**. In order to measure reversal potentials, a single channel was obtained under symmetric conditions (840 mM KCl, 500 μL in each electrophysiology chamber) and the electrodes were balanced. The 400 μL of a buffered stock solution of 3.36 M KCl was then slowly added to *cis* chamber, whereas 400 μL of salt-free buffered solution was added to the *trans* chamber to obtain a total volume of 900 μL in both sides (*trans:cis*, 467 mM KCl:1960 mM KCl). After the equilibrium was reached, IV curves were collected from −30 to +30 mV. The resulting voltage at zero current is the reversal potential (V*r*). The ion selectivity ($P_{K^+}/P_{Cl^-}$) was then calculated using the Goldman–Hodgkin–Katz equation[52], Eq. (1), where $[a_{K^+/Cl^-}]_{cis/trans}$ is the activity of the $K^+$ or $Cl^-$ in the *cis* or *trans* compartment, R the gas constant, T the temperature and F the Faraday's constant.

$$\frac{P_{K^+}}{P_{Cl^-}} = \frac{[a_{Cl^-}]_{trans} - [a_{Cl^-}]_{cis} e^{V_r F/RT}}{[a_{K^+}]_{trans} e^{V_r F/RT} - [a_{K^+}]_{cis}} \tag{1}$$

The activity of ions was calculated by multiplying the molar concentration of the ion with the mean ion activity coefficients (0.649 for 500 mM KCl, and 0.573 for 2000 mM)[56]. Ag/AgCl electrodes were surrounded by 2.5% agarose bridges containing a 2.5 M NaCl solution.

**Molecular models of type I, II, and III FraC nanopores**. The three-dimensional models with different multimeric order, ranging from five to nine monomers, were constructed with the symmetrical docking function of Rosetta[57]. A monomer without lipids was extracted from the crystal structure of FraC with lipids (PDB_ID 4tsy[33]). Symmetrical docking arranged this monomer around a central rotational axis ranging in order from 5 to 9. In total, Rosetta generated and scored 10,000 copies for each symmetry. In all cases, a multimeric organization with a symmetry similar to the crystal structure could be identified as a top scoring solution. However, in the pentameric assembly, the multimer interface was not fully satisfied as compared with the crystal structure, with large portions left exposed. The ninefold symmetric model, however, exhibited a significant drop in Rosetta score compared with the six-, seven-, and eightfold symmetric models indicating an unfavored assembly of the nonameric assembly with the six-, seven-, and eightfold assemblies as the most plausible. To create lipid-bound models, the crystal structure with lipids was superimposed on each monomer of the generated models, allowing the lipid coordinates to be transferred. The residues within 4.5 angstrom of the lipids were minimized with the Amber10 force field.

## Data availability

The authors declare that the data supporting the findings of this study are available within the article and its supplementary information files or from the corresponding authors upon reasonable request.

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

## Acknowledgements

This work is financially supported by ERC consolidator grant.

## Author contributions

G.H. and G.M. designed the experiments. G.M. supervised the project. G.H. performed the experiments and data analysis. A.V. made the molecular models. G.M. and G.H. wrote the manuscript.

## Additional information

**Competing interests:** The authors declare no competing interests.

