## [Peer Review File · Nature Communications]

Reviewers' comments:

Reviewer #1 (Remarks to the Author):

Authors have used the magnitude of residual currents made by individual peptides that translocated through FraC nanopores of varying internal sizes. The amount of collected data is laudable and experiments were conducted in a careful manner. Protein chemistry work, manipulation of current measurements, and their statistical analysis appear to be well executed. Despite these strengths, there are two major concerns of this reviewer, the novelty dimension of this piece of work and the conclusions resulting from some experiments, which cannot be generally applied to other unrelated peptide sequences. Other comments and immediate weaknesses follow.

1- It should be noted that these current blockages are not resulting from individual residue contributions, but from the overall occlusion of the ion flow. If so, there are questions associated with the origin of these current distinctions, many of which are not addressed in this manuscript.

2-Title suggests that this study aims the mass determination of individual peptides at single residue resolution. I do not see that the peptide mass is either directly or indirectly measured.

3-Single-residue resolution is accomplished just in a single case (the comparison Angio II/Angio A), but not replicated in other instances, damping the enthusiasm of reader (see comment 2). Some blockage distinctions are almost within the noise level. The overall residual currents span a certain interval (e.g., in Fig. 2, (ii) on the left panel, the second and seventh events of Angio IV show different residual currents).

4-The concept for peptide discrimination at single-amino acid resolution is quite similar to that recently published by Piguet and co-workers (2018). I mean the same current measurements, but with different pores, aiming characterizing peptides at single-residue resolution). What is fundamentally different between this work and prior work of Piguet et al. (2018), other than specifics of peptides and pores, while the approaches and aims are identical?

5-It was authors' opportunity to clarify these concerns in the discussion section, but this information is unfortunately missing. The discussion section should have included qualitative contrasts and comparisons between this work and other prevailing technologies. What was learned from this work, so other specific steps must be undertaken for unrelated peptide sequences?

6-Why are the measurements carried out far away from physiological salt conditions (1 M KCl)? Could the low signal-to-noise ratio still be mitigated by other mechanisms than very high salt concentrations.

7-In Fig. 3, what's the reason for fitting with a second-order polynomial? If so, what's the outcome of this fitting?

8-"most likely reflecting a larger overlap of the electrical double layer in the nanopores with a narrower constriction." It is not clear what is the meaning of this statement.

9-"When the four peptides were tested simultaneously, individual peptides could be readily discriminated (Fig. 2a)." This is not true. See above comment 3, in which Angio iV, for example shows a range of residual currents, given $\sim 39 \pm 4$ % for Angio IV (Fig. 2a (i)).

10-Well defined size and ion selectivity of different pores might be advantageous for some peptides, but not good traits for other peptides, as revealed in this work. This highlights tremendous difficulties for peptide/protein sequencing using small pores, primarily due to their broad range in physicochemical properties. Perhaps, coupling current measurements with another approach would drive these sorts of studies to some transformative discoveries leading to protein

sequencing.

Reviewer #2 (Remarks to the Author):

In this paper, Huang et al. engineer 3 different types of the Fragaceatoxin C biological nanopore and use them to analyze a variety (17) of short (< 22 amino acids, aa) peptides of different chemical nature, masses and electrical charge.

The 3 nanopore types differ by their ionic conductance, which is interpreted as corresponding to 3 different nanopore sizes with different number of associated monomers.

The authors show that several peptides can be distinguished based on the blockades of the nanopore ionic current they induce.

In particular, two peptides made of 8 aa and differing by a single aa (glutamic acid vs alanine) are distinguished.

The authors also show that, when using the appropriate nanopore type for the appropriate peptides, depending on the peptide molar mass, and under strictly-controlled pH conditions (pH=3,8 ; >3,6 ; <4,0), the depth of the peptide current blockade scale with the mass of the analyzed peptides, regardless of the peptide chemical composition.

The authors claim that this approach can be used as a "nanopore peptide mass identifier" which opens perspectives towards nanopore-based protein sequencing.

While several papers already explored the detection/analysis/discrimination of peptides with a biological nanopore (réfs), even at the single amino acid resolution scale, the novelty of the paper is:

- the engineering of different "sizes" of a same biological nanopore to obtain sensors of appropriate "size" for analytes of different "sizes" ; as the lack of size versatility is one of the main drawbacks of the biological nanopores compared to the artificial nanopores, this result is of significant importance
- the first demonstration of the discrimination of two peptides having the same number of aa but differing by the chemical nature of a single aa
- the wide range of the peptides analyzed compared to previous works

The work appears to be of quality and the results of the paper will certainly be of great interest to others in the nanopore community.

However in my opinion the authors should be more careful with the interpretation of the experimental results.

In particular they should avoid to generalize to all types of peptides the results obtained with a still limited range of peptides, even if wide compared to previous works, and under strictly-controlled experimental conditions.

In addition, I feel that a clarification of some presented experimental data, as well as additional experimental data, are required to strengthen the paper.

I would recommend major revisions to the manuscript before re-considering it for publication.

In addition, a careful reading of the manuscript and the correction of typographic and syntax errors are required.

Detailed comments:

Title:

Change the title.

In particular, don't use the word "identify" which suggests that unknown masses of peptides are

determined through the nanopore analysis, which is not the case.

What does "single amino chain" mean?

New title suggestion: "Engineered FraC nanopores for the analysis of oppositely-charged peptides of different masses."

Introduction:

The introduction needs significant revision to make the paper concise and understandable and less speculative (see below). I suggest the introduction to be partially rewritten in order to describe the history of the problem by focusing on recent progress in peptide sensing with single amino acid resolution (ref 14) and mass "size" sensitive peptides detection by ionic sensing (ref 14, ref 17). Ref14 and Ref 15 need to be highlighted in their precise context.

- lines 51-52: determining the mass of a peptide is not sufficient to determine its aa sequence (e.g. different orders of the same aas give the same mass; isomers ; different aa combinations providing similar masses to close to be discriminated...); does not seem to me to be the most appropriate method, or at least to be the first one to follow. In addition the term mass spectrometry may be somewhat misleading, as this technique does not discriminate molecules on the basis of their mass, but rather by their size.

- lines 81 and 82: replace "identified" with "distinguished" or with "discriminated".

- lines 84-85: really?

Results:

Global comment: results seem to have been obtained in 1M NaCl at pH=7,5 and in 1M KCl at pH=4,5 (cf legends of Supp Figs 1, 2 and 3 ; counter-example in legend of Fig 1c but is it correct?)
-> decorrelate effect of pH and salt nature?

Engineering the size of FraC nanopores

Authors used molecular modelling to predict the diameter of type II (1.1 nm) and type III (0.8 nm) FraC nanopores. As these pores are used for the first time, I think, it would have been nice if the authors had evaluated the diameters of different engineered nanopores by experiment measurements (Electron microscopy, High resolution AFM or cryo-EM reconstruction images of transmembrane pores of FraC inserted in liposomes, or PEGs partitioning?). Experiment measurements render the paper scientifically sound. Simple calculation based on measured conductance 'G' leads to obtain diameters predicted by molecular modelling. For instance

$D_{typeII}(nm) = D_{typeI} * (G_{typeII}/G_{typeI})^{0.5} = 1.6 * (1.08/2.07)^{0.5} = 1.1$; modelling simulation seems too good. Please comment on it.

Fig 1:

a) a schematics of the experimental configuration is missing

b) proportion of the different pore types vs pore version and pH:

Supp Fig 3 clearly explains how the results shown on Fig 1b are obtained.

Call Supp Fig 3 in the text earlier than in line 109, e.g. around lines 94-95.

c) as peptide experiments have essentially been performed at pH=4,5, it would have been interesting to show IV curves at pH=4,5 instead of pH=7,5.

Legend: is it really 1M KCl? other experiments at pH=7,5 have been performed in 1M NaCl (see legends of Supp Figs 1, 2 and 3).

It would also have been interesting to show IV curves of the 3 pore types of a given pore version at pH=4,5

d-e) indicate in the figure and in the legend that these measurements are performed at -50 mV. Compare the type III WT-FraC conductance at pH=4,5 to the type I and type II WT-FraC conductance at pH=4,5 in the text (line 97).

Highlight the variation of the pore current noise between the different pore types in the text. These 2 points can also be discussed around line 120.

Identification of single amino acid substitutions with type II FraC nanopores

angiotensins discrimination:

Fig 2:

b) iii-iv) the Ires% value of the maximum of the AngioII histogram falls between (roughly at equal distance) of the Ires% values of the 2 maxima of the AngioII+AngioA mixture histogram.

In addition, as the S.D. values of the two maxima of the mixture histogram are very close, this questions the identification of the AngioII vs AngioA populations.

Origin of the shift?

Reproducibility?

What is measured for AngioA alone?

This is of particular importance as the distinction of these 2 peptides differing by a single aa is one of the main claims of the paper.

smaller peptide differences:

lines 147-149 of the main text: "smaller peptide differences, e.g. 34 Da difference between phenylalanine Phe7 in angiotensin III and Ile7 in angiotensin III, were observed but not easily detected".

Supp Fig 6: I agree with the authors that the difference is not easily observed.

What is observed if the same number of blockades is represented in b and c histograms?

Are n=227 or 401 blockades not too small numbers of blockades to obtain reproducible results?

Would a blockade duration vs Ires% histogram help to discriminate the populations as the mean blockade durations are 0,35 ms vs 0,19 ms?

peptides analyzed with type III nanopores:

lines 150-152 of main text: it would have been interesting to show current traces of these peptides with type II nanopores as a Supp Fig to illustrate the lack of resolution

Supp Fig 7 c): Met5-enkephalin is missing.

Compare with histograms of independent experiments with peptides of each given chemical nature.

In particular the signal attributed to Leu-enkephalin is weak compared to the other signals.

Supp Fig 8: Angiotensin IV is missing.

2 Ires% levels for endomorphin I: the authors exclude level 1 and keep level 2 based on the following arguments:

- level 1 blockades are more longer (1 order of magnitude) than for other peptides while level 2 blockades have durations similar to other peptides ; so?

- level 1 Ires% constant with voltage while level 2 Ires% increase with voltage like for other peptides: not clear from Supp Fig 11 ; we don't have comparison with other peptides

- level 1 blockade duration constant with voltage while level 2 blockade duration decrease with voltage like for other peptides: ok from Supp Fig 11 (but we don't have comparison with other peptides)

- error bars on Supp Fig 11?

- the authors interpret level 1 as peptide dimers and level 2 as peptide monomers: use reducing

agent to confirm this hypothesis?

A nanopore mass spectrometer for peptides

Fig 3:

- type I from WT-Frac, type II from W116S-FraC, type III from W112S-W116S-FraC: why not the 3 pore types from a same pore version (e.g. from W112S-W116S-FraC which provide similar proportions of the 3 pore types at pH=4,5)?
- type I and type II at -30 mV but type III at -50 mV: voltage effect on Ires% vs peptide mass?
- b) Angiotensin IV is missing
- the "deviating" c-Myc peptide is the only negatively charged peptide at pH=4,5 ; it would have been of great interest to test other negatively charged peptides to see if the electrical charge is a critical parameter
- what are the results of Fig 3 at pH=7,5 (more diversity in peptide charges than at pH=4,5)? use D109S-W116S FraC which provides the 3 nanopore types at pH=7,5 and 4,5 (see Supp Fig 2h and Supp Fig 3f) but does not seem to be exploited ; e.g. comparing type II nanopore analysis of the same peptides at pH=7,5 and at pH=4,5 could be of particular interest.

Supp Fig 9: Angiotensin I is missing

Fig 4:

need to be in very particular and strictly-controlled pH conditions (pH=3,8 ; >3,6 ; <4,0) for c-Myc to follow the Ires% vs peptide mass tendency of the other peptides

- b) what is the Ires% vs pH behaviour of the other peptides analyzed with type II nanopore (only 4 over 9 are shown)?
- c) 2 peptides are missing
- d) non monotonic behaviour of blockade duration of c-Myc vs pH for the different voltages ; maximum reached at pH=3,8 ; in addition to the need of strictly-controlled pH conditions, this suggests that pH=3,8 is a very particular pH value for this peptide, which creates doubt about the possible generalization of the technique ; testing other negatively charged peptides is of crucial importance as the main claim of the paper is the use of the system as a nanopore mass spectrometer

Discussion:

lines 204-205: these are calculated values ; need to be measured experimentally

lines 248-250: too much generalization regarding the results.

Reviewer #3 (Remarks to the Author):

In this manuscript, Huang et al. demonstrate FraC pores can be engineered to three sizes, with the relative yield of each size-type being tuneable by varying synthesis conditions. The authors then characterize mixtures of short (4 - 22 AA) peptides with improved sensitivity due to the longer dwell times and enhanced volumetric sensitivity elicited by the smaller (type II) pores. Methods to improve the yield of these smaller pores to appreciable percentage levels (e.g. from c.a. 0-3% to 14-20% for type III pores from WT-FraC) are presented. These include varying pH, the choice of chemical substitution at the lipid interface, elution, the concentration of monomers during liposomes and optimization of the purification protocol.

The authors then identify small volume differences (c.a. 40 Da) in mixtures of translocating peptides using the nanopores.

Importantly, under the right conditions, the signal appears proportional to the mass despite the fact that some substitutions have 'opposite charge'. Taken cumulatively, these findings are actually quite significant with respect to the literature; the production of sub-nm biological pores alone is novel. Being capable of identifying residue-scale volume differences in protein/peptides irrespective of charge is even more so. While there is a lot of pH dependent nanopore data in the literature, Figure 4 b and c are unusually striking examples of specificity. Overall, I recommend the paper for publication with minor corrections to be addressed below:

General issues:

1. One issue is the question of robustness and longevity of the pores given the embedding protocol includes a (presumably weaker) lipid interface serine bond among other changes such as pH etc. Please provide a comment (or ideally quantitative data) on the lifetime of pores broken out by type (I,II,III). Even if the types are equivalently functional, this would be instructive to the reader.

2. Using pores with sub-nm or nearly nm cross section is expected to drastically increase clogging during translocation – a comment on the authors experience with the smaller pores would be useful. Potentially related to this, why were type II pores used predominantly for translocation experiments since 0.8 nm type I pores would give much more sensitivity to smaller peptides? Would this require too much bias to achieve translocation?

3. It is not clear how many blockades are required in order to discriminate at 40 Da glutamate/alanine resolution (1,10,100 blockades?). A more statistically robust assertion of volumetric discrimination as a function of blockade count could strengthen the paper. A trend of 'volume sensitivity (Da)' plotted as a function of blockade count for the different pore types found using a bootstrapping approach would add rigor to the claim of volume sensitivity. (One option would be to take N blockades, say 5 randomly chosen blockades of type alanine vs 5 others and find what Da resolution on average is achieved, then repeat for N = 1...1000 etc and plot against N - the blockade count). The trend should limit to say, 40 Da passed some sufficient number of molecules.

4. It is also unclear how this work could be extended or enhanced - the authors commendably found small anomalies in their data (such as a <1% yield of 0.8 nm pores) and exploited it – but is there a way this could be extended to even smaller pores or is this a physical limit of the structure? Looking at the trend of smaller pore yield as a function of conditions, I initially wondered if the authors reduced the concentration of monomers during liposomes even further if yields could improve, but as in Figure S.1, a 1:80 protein:lipid ratio did not give nanopores in lipid bilayers. Did the authors ever observe even smaller pores. Is there an anticipated avenue for further engineering? Please add a discussion comment.

5. The authors spend a lot of time saying mass spectrometry is limited in various ways "extremely complex and unwieldy, and are not single-molecule." The authors spend considerable time framing the work within the context of mass spectrometry so the point is salient, but I would argue that the 1 second acquisition required by a mass spectrometer to get a 0.001 Da accuracy mass identification outweighs the limits the authors put forward. There are actually a lot of synergies between mass spectrometry and nanopore sequencing. The nanopore mass spectrometer devices produced by Derek Stein are an excellent example of the potential complementarity between these technologies. The paper might benefit from a more complimentary introduction and discussion of the respective technologies.

Minor issues:

1. Claiming that the smaller pores were the 'dominant species' yield is somewhat unfair when both

type II and III have to be grouped for this statement (line 108) – please rephrase (according to the supplement the increase was c.a. 0-3% to 14-20% for type III pores, but moreso in other types – the reality is that results appear mixed and specific)

2. In Figure 2, the authors have interestingly plotted Ires against S.D. because it can discriminate peptides – but why? Why does standard deviation permit distinction of species? Obviously translocation dynamics are the cause but in what sense? For example the angio II shows a larger standard deviation than angio I despite its smaller size. But then, angio IV (the smallest) inverts this trend by also having the smallest S.D current on average. Please explain or provide an intuitive explanation or justification for this in the text.

3. Comparing Figure 2 to the state of the art, I note that the Wanunu lab “Identification of amino acids with sensitive nanoporous MoS₂: towards machine learning-based prediction npj 2D Materials and Applications (2018) 2: 14” and Timp lab “Reading the primary structure of a protein with 0.07 nm³ resolution using a subnanometre-diameter pore - Nature nanotechnology 11 (11), 968 2016”, and Pevzner lab “Single-molecule protein identification by sub-nanopore sensors PLoS computational biology 13 (5), 2017” have separately found random forest regression to be the superior approach for classification of peptides blockades in sub-nanopores. Even if the authors do not want to perform a more complex analysis on their peptide blockades, I highly encourage these references be included in a statement that more complex classification of peptides has been demonstrated elsewhere, and would likely improve the sensitivity of discrimination - but S.D. and Ires are sufficient here for a proof of principle demonstration.

4. The figure 2 surface plots require attention; please label the color bar axes, e.g. as ‘count’. Do the authors mean 0 to be white in the multi-colored label? Currently, ‘0’ is denoted a dark blue color in the color label, but no dark blue is present in the map despite 0 being present across most of the space. This is confusing when compared to Supplementary Figure 6 where the background is actually blue – why? Contour plots are not as illustrative as kernel density maps for showing non-uniform distributions, but it’s a superficial point but I encourage Figure 2, S6 and S7 to be redrawn.

5. Continuing with Figure 2 iii-iv, the labels iii and iv are both used twice – is this intentional? If so, group the plots together and remove one label. The maps show different numbers of counts and different numbers in total – please normalize the map color height in some intuitive way (e.g. setting the max to 50, or to some fraction of total counts in each case).

6. It is hard to compare all 4 Figure 2 maps directly because the axes are not the same – please set the x axis (e.g. ranging from 0 to 45%) as constant in each case (instead of 2, 2). Consider logarithmic scales if it improves visualization and in the supplement.

7. ‘Current blockade’ can refer to the magnitude of current associated with the blockade of a pore, or to the absolute level of a pore when blocked. On lines 134 – 135, Ires%, is defined as $IB / IO \times 100$, but if it was current magnitude then the residual current would be defined as $(1 - IB / IO) \times 100$. I recommend IB be more precisely defined as the ionic current level during a blockade.

8. The same peptide will exhibit a different Ires% in different sized pores, because it will occupy more/less volume. However, across those same pores, it will have same absolute current blockade magnitude provided the bias/electrolyte is constant. Therefore why is this statement true on line 134 that “Residual currents percent (Ires%, defined as $IB / IO \times 100$) were used instead of current blockades because they provided more reliable values when comparing different nanopores.”? Wouldn’t blockade magnitude, $|IB - IO|$, be superior across pores of different size because it is not sensitive to pore size?

9. In Figure 3, the residual current is broken out as a function of molecular weight. There is clearly a relationship between size and residual current. However, what is the purpose of performing a

polynomial fit since the actual pore and peptide size are known as well as the translocation bias? Therefore the expected conductance, G can be explicitly determined and the peptide radii could be coarsely estimated as $V(\text{nm}^3) = 1.212 \times 10^{-3} (\text{nm}^3/\text{Da}) \times \text{MW}(\text{Da})$ as one example from which blockade fraction is easily estimated. An explicit fit or simple analytical expression using the known parameters should at least be attempted, as it would add further corroboration of the pore size if it fit the experimental data instead of just a guide to the eye.

10. It would improve the paper to add an analysis section to the methods and provide details like the filter settings in one place. The use of a 10 kHz is repeated in the Figure 1, Figure 2 and Figure 4 captions and again in table 1 and then again multiple times in the supplement. This seems wasteful since the vast majority of experiments were done in the same buffer at the same bias and settings. The manuscript would be improved if the authors gathered comprehensive details about how the data was handled, culled, cleaned, processed and plotted into the analysis section. While the filter used is relevant, it would be far more instructive to know how data was handled (manual cropping of current trace, one user/multiple users/repeatability?) and how specifically the tables were arrived at.

11. Please check the manuscript for typographical errors, e.g. "Bassel filter" should read "Bessel filter"

12. Supplementary Figure 9 caption – it is not amplitude it is residual blockade current fraction.

13. In supplementary Figure 8, Endomorphin I is badly fit since the Ires distribution is multimodal – is the secondary presence of many near 0% Ires measurements at long dwell times due to aggregates/clogging? The authors discussed this but do not attribute a cause or solution. Consider repeating the experiment with fresh, reduced analyte concentration. Figure. S11 appear to suggest this as being a multilevel blockade phenomena – if so please add a few raw events current traces showing multiple levels with level guidelines to Figure S11.

Reviewers' comments:

Reviewer #1 (Remarks to the Author):

Authors have used the magnitude of residual currents made by individual peptides that translocated through FraC nanopores of varying internal sizes. The amount of collected data is laudable and experiments were conducted in a careful manner. Protein chemistry work, manipulation of current measurements, and their statistical analysis appear to be well executed. Despite these strengths, there are two major concerns of this reviewer, the novelty dimension of this piece of work and the conclusions resulting from some experiments, which cannot be generally applied to other unrelated peptide sequences. Other comments and immediate weaknesses follow.

We thank the reviewer for the comments. Concerning for the novelty of this work, we believe this work describes major steps towards single-molecules peptide mass identification using nanopores, and describes a new approach for real-time protein sequencing. As a more neutral assessment for the novelty of this work we paste below the points highlighted by reviewers 2 and 3:

- The isolation of a sub-nm biological pores, that is the smallest biological nanopore known to date
- The ability of the nanopores to identify peptides on volume(mass) differences irrespective of charge
- The recognition of differences in peptides that have the same number of residues but differ by one amino acid
- The wide range of the peptides analyzed compared to previous works

As for the other comments:

1- It should be noted that these current blockages are not resulting from individual residue contributions, but from the overall occlusion of the ion flow. If so, there are questions associated with the origin of these current distinctions, many of which are not addressed in this manuscript.

We are not sure what is the reviewer asking. Over the past two decades, hundreds of manuscripts showed that current blockades are from individual analytes. Hence, we are pretty

confident the current blockades described here are also from individual analytes. Another possible interpretation of the reviewer comment is that he/she is disappointed that we do not sequence peptides. If this is the issue, indeed, we do not sequence individual peptides. We think we made it quite clear that we identify the volume (mass) of the peptides, not their sequence. Here we describe the equivalent of a nanopore mass spectrometer. As in tandem mass spectrometry, the mass of the peptides can also lead to their sequence, providing several other steps are taken. At the end of the manuscript we have discussed an approach that would allow using the nanopores described here to sequence proteins using our approach.

2-Title suggests that this study aims the mass determination of individual peptides at single residue resolution. I do not see that the peptide mass is either directly or indirectly measured. In order to have a nanopore peptide mass identifier, the ionic current signal of individual peptides must show a direct correlation with their mass. Figure 3 (now Figure 4) showed that the residual excluded current of 17 peptides is directly correlated to their volume (mass). Exceptions were two negatively charged peptide. Figure 5 (old Figure 4) shows that lowering the pH to 3.8, all peptides, including the negatively charged ones, showed a direct correlation between their residual current and their mass. These experiments, therefore, show that by measuring the ionic blockade of an unknown peptide it will be possible to know its mass (volume).

3-Single-residue resolution is accomplished just in a single case (the comparison Angio II/Angio A), but not replicated in other instances, damping the enthusiasm of reader (see comment 2). Some blockage distinctions are almost within the noise level. The overall residual currents span a certain interval (e.g., in Fig. 2, (ii) on the left panel, the second and seventh events of Angio IV show different residual currents).

We tested several different peptides differing by one single-amino acid, using type I, type II and type III nanopores. For example, Figure 2 shows discrimination between angiotensin II and angiotensin III, differing by one single amino acid. Figure 2 also shows that angiotensin III and angiotensin IV, which different by one amino acid, can be distinguished. Figure S10 shows discrimination between four peptides (angiotensin IV, angiotensin 4-8, leu-enkephalin and endomorphin I), all differing by one amino acid. Finally, we are the first to show the separation of two peptides having the same number of residues but differing by one single amino acid.

The reviewer is correct that some blockades overlap. We are not claiming this approach can identify a peptide mass from a single blockade. Multiple blockades should be analysed.

4-The concept for peptide discrimination at single-amino acid resolution is quite similar to that recently published by Piguet and co-workers (2018). I mean the same current measurements, but with different pores, aiming characterizing peptides at single-residue resolution). What is fundamentally different between this work and prior work of Piguet et al. (2018), other than specifics of peptides and pores, while the approaches and aims are identical?

We are aware of this work, which was cited and discussed. There are crucial differences between the work of Piguet *et al* and this work.

- 1) Piguet *et al* used model peptides, bearing a uniform positive charge (poly-arginine or poly-lysine residues) differing by one amino acid in length. In this work we moved away from model analytes and used real-world peptide samples. This is a crucial step, because until now, it was not clear whether peptides with different charges could be sampled by nanopores.
- 2) In the above mentioned work the two peptides differed by 146 Da, while here we show differences of 44 Da.
- 3) We measured peptides with the same number of residues but differing by one amino acid.
- 4) Crucially, we also demonstrated that under selected conditions (*i.e.* pH 3.8) the peptide blockades scaled directly with the volume(mass) of the peptide despite the analyte chemical composition. This finding, was not expected and suggests this approach can be used to measure the mass of any peptide.

5-It was authors' opportunity to clarify these concerns in the discussion section, but this information is unfortunately missing. The discussion section should have included qualitative contrasts and comparisons between this work and other prevailing technologies. What was learned from this work, so other specific steps must be undertaken for unrelated peptide sequences?

Following the reviewer criticisms, we have clarified our interpretation to the data. We believe we have explained that the mass of unrelated peptides can be measured using our approach. We

also have re-written the introduction to accommodate the discussion of previous technology to recognize and sequence proteins and peptides at the single-molecule level.

6-Why are the measurements carried out far away from physiological salt conditions (1 M KCl)? Could the low signal-to-noise ratio still be mitigated by other mechanisms than very high salt concentrations.

The ionic strength of the measurements is an arbitrary choice. 1 M KCl is often used in nanopore recordings because it gives high signal to noise ratio. However, higher or lower ionic strengths can be used.

7-In Fig. 3, what's the reason for fitting with a second-order polynomial? If so, what's the outcome of this fitting?

The data should be included between zero (empty pore) and one (a fully occupied pore). We tested several empirical fittings and only a polynomial fitted well the experimental data. Most likely, the capture and immobilization of peptides depends on several components. We can give the values of the fittings, but we think it would be meaningless without a physical explanation of the process. We prefer providing the extrapolated value for the fully occupied pore, which is important to estimate the recognition volume inside the nanopore, and the raw data. The latter can be used by other groups in more in-depth biophysical investigations.

8-“most likely reflecting a larger overlap of the electrical double layer in the nanopores with a narrower constriction.” It is not clear what is the meaning of this statement.

When a charged surface is immersed in an electrolyte solution, a layer of charges will line the surface, which is called electrical (Helmholz) double layer. Ions diffusing near the surface (Gouy-Chapman diffusive layer) induce an electrical potential that exponentially decays moving away from the charged surface. The extension of such potential depends on the Debye length, which is typically between 0.2 and 1 nm. In nanopores, the distance between the inner walls can be smaller than the Gouy-Chapman diffusive layer, providing a potential overlap.

9-“When the four peptides were tested simultaneously, individual peptides could be readily discriminated (Fig. 2a).” This is not true. See above comment 3, in which Angio iV, for example shows a range of residual currents, given $\sim 39 \pm 4$ % for Angio IV (Fig. 2a (i)).

We are not claiming our approach is perfect, and we are not denying there is some overlap between the signal of different peptides. Part of the aim of this work is to define the limitations of the nanopore mass spectrometer. As shown in Figure S7, differences in 34 Da cannot be easily observed (using some more sophisticated data analysis software can actually distinguish smaller differences). Nonetheless, we think that differences between the four peptides shown in figure 2 can be observed, despite some signal overlap.

10-Well defined size and ion selectivity of different pores might be advantageous for some peptides, but not good traits for other peptides, as revealed in this work. This highlights tremendous difficulties for peptide/protein sequencing using small pores, primarily due to their broad range in physicochemical properties. Perhaps, coupling current measurements with another approach would drive these sorts of studies to some transformative discoveries leading to protein sequencing.

Nanopores with well-defined structures are of paramount importance. Hence, the ability of preparing nanopores holding different sizes or geometries is extremely useful, if we ought to detect a wide range of analytes. Peptides have different sizes, so it is likely that nanopore with different sizes will be required to analyse a wide range of peptide masses.

Here we showed that at pH 3.8 the mass of peptides can be identified using nanopores. However, the sequence of the peptide cannot be identified. At the end of the manuscript we have discussed an approach that would allow using the nanopores described here to sequence proteins.

Coupling nanopore measurements with other techniques is challenging, but indeed might be advantageous. We have discussed now in the introduction an approach that couples nanopores with conventional mass spectrometry.

Reviewer #2 (Remarks to the Author):

In this paper, Huang et al. engineer 3 different types of the Fragaceatoxin C biological nanopore and use them to analyze a variety (17) of short (< 22 amino acids, aa) peptides of different chemical nature, masses and electrical charge.

The 3 nanopore types differ by their ionic conductance, which is interpreted as corresponding to 3 different nanopore sizes with different number of associated monomers.

The authors show that several peptides can be distinguished based on the blockades of the

nanopore ionic current they induce.

In particular, two peptides made of 8 aa and differing by a single aa (glutamic acid vs alanine) are distinguished.

The authors also show that, when using the appropriate nanopore type for the appropriate peptides, depending on the peptide molar mass, and under strictly-controlled pH conditions (pH=3,8 ; >3,6 ; <4,0), the depth of the peptide current blockade scale with the mass of the analyzed peptides, regardless of the peptide chemical composition.

The authors claim that this approach can be used as a "nanopore peptide mass identifier" which opens perspectives towards nanopore-based protein sequencing.

While several papers already explored the detection/analysis/discrimination of peptides with a biological nanopore (réfs), even at the single amino acid resolution scale, the novelty of the paper is:

- the engineering of different "sizes" of a same biological nanopore to obtain sensors of appropriate "size" for analytes of different "sizes" ; as the lack of size versatility is one of the main drawbacks of the biological nanopores compared to the artificial nanopores, this result is of significant importance
- the first demonstration of the discrimination of two peptides having the same number of aa but differing by the chemical nature of a single aa
- the wide range of the peptides analyzed compared to previous works

The work appears to be of quality and the results of the paper will certainly be of great interest to others in the nanopore community. However in my opinion the authors should be more careful with the interpretation of the experimental results.

In particular they should avoid to generalize to all types of peptides the results obtained with a still limited range of peptides, even if wide compared to previous works, and under strictly-controlled experimental conditions.

In addition, I feel that a clarification of some presented experimental data, as well as additional experimental data, are required to strengthen the paper.

I would recommend major revisions to the manuscript before re-considering it for publication.

In addition, a careful reading of the manuscript and the correction of typographic and syntax errors are required.

We thank the reviewer for the comments. We appreciate the nice summary of the advantages of this work.

Detailed comments:

Title:

Change the title.

In particular, don't use the word "identify" which suggests that unknown masses of peptides are determined through the nanopore analysis, which is not the case.

What does "single amino chain" mean?

New title suggestion: "Engineered FraC nanopores for the analysis of oppositely-charged peptides of different masses."

The title is attempting to summarize the findings in this manuscript. The main findings we would like to highlight are:

- We can engineer the size of the nanopore
- Nanopores can analyze peptides with opposite charge
- There is a relation between the current blockades and the volume (mass) of the analyte
- We can observe differences in peptides bearing the same number of amino acids, but differences of one side-chain

According to the reviewer, these are also the important points in our manuscript.

Hence, taking into consideration the comment of the reviewer, we would like to use:

FraC Nanopores with Adjustable Diameter Identify the Mass of Opposite-Charge Peptides with 44 Da Resolution.

We decided to add the reference to the engineered diameter, as it is one of the main findings of this work. We also kept the term "identify", because there is a direct correlation between the mass of the peptide and its blocked ionic current. Hence, the mass of peptides can be identified without a prior knowledge of the peptide itself, i.e. peptides can be *identified*. The term: "single amino chain" in the original title meant to refer to peptides containing the same length but just differ in one amino acid. However, we substituted single amino chain resolution with the actual mass difference between the two peptides.

We would like to stress that one of the main discoveries of this work is that under carefully selected conditions (*i.e.* pH 3.8) the mass (volume) of the peptide scales directly with the depth of ionic current blockade. We find this concept important because it allows the nanopore to

collect information about the analyte without prior knowledge of the analyte itself. Previous work with peptides did not tackle this issue. Our finding is also rather surprising, because work with DNA and other molecules suggested that the chemical identity of the analyte has an influence on the ionic signal (i.e. the blockades inferred by different nucleobases do not scale with their molecular mass). Hence our nanopore approach allows the *identification* of peptide masses. This is the reason why we have kept the term “identify” throughout the manuscript.

Introduction:

The introduction needs significant revision to make the paper concise and understandable and less speculative (see below). I suggest the introduction to be partially rewritten in order to describe the history of the problem by focusing on recent progress in peptide sensing with single amino acid resolution (ref 14) and mass “size” sensitive peptides detection by ionic sensing (ref 14, ref 17).

Ref14 and Ref 15 need to be highlighted in their precise context.

As suggested by the reviewers, we have extended the description of prior art.

- lines 51-52: determining the mass of a peptide is not sufficient to determine its aa sequence (e.g. different orders of the same aas give the same mass; isomers ; different aa combinations providing similar masses to close to be discriminated...); does not seem to me to be the most appropriate method, or at least to be the first one to follow. In addition the term mass spectrometry may be somewhat misleading, as this technique does not discriminate molecules on the basis of their mass, but rather by their size.

Indeed, we are not sequencing individual peptides, but we identify their mass. We have rewritten the introduction and explained more in detail what we meant. Today mass spectrometry is the technique of choice to sequence proteins, hence it is possible to sequence proteins by measuring the mass of individual peptides. In the introduction we have drawn a parallel between conventional mass spectrometry and nanopore mass-identifier. Despite many groups have been using the term nanopore mass spectrometry, we are aware this term might not be appropriate. Hence, we took care not to using it here. We have called our system a mass-identifier. We are also not claiming this strategy is immediately applicable to protein sequencing. This technology can measure the mass of peptides, which in turn can be used to sequence proteins, in the same way a mass spectrometer is now the method of choice to sequence proteins.

However, the reviewer is correct, the nanopore is identifying the analyte based on his volume. Nonetheless, there is a direct correlation between the volume of a peptide and its mass. We have re-written part of the manuscript to make this point clearer.

- lines 81 and 82: replace "identified" with "distinguished" or with "discriminated".

In this manuscript we establish a direct correlation between the intensity of the ionic signal and the mass (or volume) of the peptide. Hence, even without knowing the analyte, its volume (or mass) can be "identified". This is why we used the term identified. If such correlation between mass and signal could not be established, then we would agree with the reviewer that the analyte could just be distinguished (from other analytes). We have tested many conditions to establish this mass vs signal correlation and this is the main message of this work, hence we decided to leave the term "identified".

- lines 84-85: really?

We propose a new method to sequence proteins in which a peptidase is attached atop of a FraC nanopore and the mass of the excised peptide is identified. To the best of our knowledge such method has not been suggested before. This concept was explained at the end of the manuscript. Now we have moved this part in the introduction and described it more in details.

Results:

Global comment: results seem to have been obtained in 1M NaCl at pH=7,5 and in 1M KCl at pH=4,5 (cf legends of Supp Figs 1, 2 and 3 ; counter-example in legend of Fig 1c but is it correct?) -> decorrelate effect of pH and salt nature?

The reviewer is correct. The nanopores were tested in 1 M NaCl at pH 7.5 and then in 1 M KCl solutions at pH 4.5. All electrophysiology experiments were carried out in K⁺ solutions because K⁺ has higher mobility than Na⁺, which in turn provides higher conductance. Since all electrical recordings were carried out at pH 4.5, (at pH 7.5 the peptides are not captured by the nanopore), at such pH we switched to a K⁺ solution. Type I, II and III nanopores can be formed in both NaCl and KCl.

Engineering the size of FraC nanopores

Authors used molecular modelling to predict the diameter of type II (1.1 nm) and type III (0.8 nm) FraC nanopores. As these pores are used for the first time, I think, it would have been nice

if the authors had evaluated the diameters of different engineered nanopores by experiment measurements (Electron microscopy, High resolution AFM or cryo-EM reconstruction images of transmembrane pores of FraC inserted in liposomes, or PEGs partitioning?). Experiment measurements render the paper scientifically sound. Simple calculation based on measured conductance 'G' leads to obtain diameters predicted by molecular modelling. For instance $D_{typeII}(nm) = D_{typeI} * (G_{typeII}/G_{typeI})^{0.5} = 1.6 * (1.08/2.07)^{0.5} = 1.1$; modelling simulation seems too good. Please comment on it.

We thank the reviewer for the suggestions. We have used the equation suggested by the reviewer and found that the calculated pore size corresponded well to the value measured from our models. We have now mentioned this in the main text and in Figure S3. We have also performed negative stain EM (Figure R1). We could observe nanopores with different sizes. However, since the nanopores are relatively small, it was not possible to infer about their exact size.

Although EM supports our interpretation, we would like to stress that although knowing the exact size / shape / structure of the nanopore would be nice, it is actually not crucial for the aim of this work. The different nanopores can be separated, and they can be used to recognize different peptides. Even if they would be FraC nanopores with identical number of subunit and different transmembrane topologies, they would still recognize the peptides.

Fig 1:

a) a schematics of the experimental configuration is missing

We are not sure what is the reviewer asking. We have added a schematic showing that an applied potential is applied across the nanopore.

b) proportion of the different pore types vs pore version and pH:

Supp Fig 3 clearly explains how the results shown on Fig 1b are obtained.

Call Supp Fig 3 in the text earlier than in line 109, e.g. around lines 94-95.

Figure S3 describes the characterization of the different nanopores including the mutant pores.

We refer to figure S3 as soon as we describe the mutants in the main text.

c) as peptide experiments have essentially been performed at pH=4,5, it would have been interesting to show IV curves at pH=4,5 instead of pH=7,5.

We have changed the IV curves

Legend: is it really 1M KCl? other experiments at pH=7,5 have been performed in 1M NaCl (see legends of Supp Figs 1, 2 and 3).

Indeed the salt was 1 M NaCl. However, as suggested by the reviewer in the next point, we have now add the IV, which were collected at pH 4.5.

It would also have been interesting to show IV curves of the 3 pore types of a given pore version at pH=4,5

We have added the IV curves as suggested by the reviewer in Fig. S4.

d-e) indicate in the figure and in the legend that these measurements are performed at -50 mV. Compare the type III WT-FraC conductance at pH=4,5 to the type I and type II WT-FraC conductance at pH=4,5 in the text (line 97).

We have done this.

Highlight the variation of the pore current noise between the different pore types in the text. These 2 points can also be discussed around line 120.

We have added a power spectrum of the pores (Figure S5) and commentated in the main text.

Identification of single amino acid substitutions with type II FraC nanopores

angiotenins discrimination:

Fig 2:

b) iii-iv) the Ires% value of the maximum of the AngioII histogram falls between (roughly at equal distance) of the Ires% values of the 2 maxima of the AngioII+AngioA mixture histogram.

In addition, as the S.D. values of the two maxima of the mixture histogram are very close, this questions the identification of the AngioII vs AngioA populations.

Origin of the shift?

Reproducibility?

What is measured for AngioA alone?

This is of particular importance as the distinction of these 2 peptides differing by a single aa is one of the main claims of the paper.

We thank the reviewer for spotting this. Actually, the blockades of AngioA and AngioII are very reproducible (Fig. R2). However, the reviewer comment made us realized it might be important for the reader to clearly see the separation between peptides. Hence, Figure 2 is now separated into two separate figures (now Fig. 2 and Fig. 3), each including the contour plots of individual peptides.

lines 147-149 of the main text: "smaller peptide differences, e.g. 34 Da difference between phenylalanine Phe7 in angiotensin III and Ile7 in angiotensin III, were observed but not easily detected".

Supp Fig 6: I agree with the authors that the difference is not easily observed.

What is observed if the same number of blockades is represented in b and c histograms?

Are n=227 or 401 blockades not too small numbers of blockades to obtain reproducible results?

Would a blockade duration vs Ires% histogram help to discriminate the populations as the mean blockade durations are 0,35 ms vs 0,19 ms?

When measuring only one analyte, thousands of data points are not an issue. However, when making these plots, we found it is more instructive to show hundreds of events rather than thousands of events. This is because the overlap of the data points between the two peptides will make the figure too crowded. Further, small changes in the baseline will lead to smearing of the signal and additional overlaps. Both the dwell time and residual current can be accurately sampled with just hundreds of events. This is shown in Figure R3 where the separation has not improved having 200 hundred or one thousands of events.

We tested representations of Ires vs dwell times, but we found that the SD of the signal gives better separation. Possibly, this is because dwell times have an exponential distribution and the peptides have a too much similar value.

lines 150-152 of main text: it would have been interesting to show current traces of these peptides with type II nanopores as a Supp Fig to illustrate the lack of resolution

We have now added a new figure in SI (Figure S6, S10) showing angiotensin I with type I pore, and angiotensin IV with type II pore.

Supp Fig 7 c): Met5-enkephalin is missing.

Compare with histograms of independent experiments with peptides of each given chemical nature.

We did not add Met5-enkephalin because the signal of Met5-enkephalin overlaps with that of leu-enkephalin and cannot be distinguished. The mass difference between Met5-enkephalin and leu-enkephalin is just 18 Da, which is beyond the resolution of our system at the moment.

Supp Fig 8: Angiotensin IV is missing.

We added the missing peptide.

2 Ires% levels for endomorphin I: the authors exclude level 1 and keep level 2 based on the following arguments:

- level 1 blockades are more longer (1 order of magnitude) than for other peptides while level 2 blockades have durations similar to other peptides ; so?

Surprisingly, the events induced by endomorphin I showed two kind of blockades. One explanation is that they are due to monomer and dimer blockades. Alternatively, they may originate from the sampling of unfolded and folded peptides. We found the latter explanation unlikely, since these peptides should be too small to be folded, especially so at low pH.

Therefore, the deep blockade might arise from dimer peptides. In this case the dimers are expected to provide deeper blocks and a slower translocation time, as observed here.

Noticeably, endomorphin I contains a proline residue and three aromatic amino acids. Similar groups of amino acids have been shown to form π - π interaction and promote the assembly of supramolecular configurations (e.g.: Brown N, et al. ACS Nano, 2018, 12 (4), pp 3253–3262 and many others).

- level 1 Ires% constant with voltage while level 2 Ires% increase with voltage like for other peptides: not clear from Supp Fig 11 ; we don't have comparison with other peptides

- level 1 blockade duration constant with voltage while level 2 blockade duration decrease with voltage like for other peptides: ok from Supp Fig 11 (but we don't have comparison with other peptides)

- error bars on Supp Fig 11?

We agree with the reviewer the correlation is weak. Hence, we decided to remove the voltage dependency for this peptide, as it does not really explain the nature of this blockade. Instead, we added the detailed characterization of the two-level blockades (lex%, dwell time, plotting of dwell time over lex% and raw traces) in new Fig S13. This shows the two events can be separated.

- the authors interpret level 1 as peptide dimers and level 2 as peptide monomers: use reducing agent to confirm this hypothesis?

The peptides have no cysteine residue and cannot make any disulfide bond, hence reducing agents is not expected to show any effect.

A nanopore mass spectrometer for peptides

Fig 3:

- type I from WT-Frac, type II from W116S-FraC, type III from W112S-W116S-FraC: why not the 3 pore types from a same pore version (e.g. from W112S-W116S-FraC which provide similar proportions of the 3 pore types at pH=4,5)?

As shown in figure S3 and explained in the text, the three pore types have the same conductance, hence they are very likely to have the same size and shape. This is expected, because the modifications are made on the outside of the nanopore. We chose to work with different mutants for practical reasons. Wt-FraC gives most type I pore, W116S-FraC gives most type II pore and W112S-W116S-FraC most of type III pores, hence we used these respective versions when we needed the different nanopores.

- type I and type II at -30 mV but type III at -50 mV: voltage effect on Ires% vs peptide mass?

We didn't measure the effect of the voltage on the Ires. We chose the external bias depending on the translocation time of analyte and we kept it constant for each nanopore type. In particular, we used -50 mV for type III pore to obtain a better signal to noise ratio. However, the Ires% include the ratio between the blocked pore and the open pore current, which should not be voltage dependent.

- b) Angiotensin IV is missing

At pH 4.5 we initially removed the data for angiotensin IV because the translocation was very fast and we were not sure whether the peptide was sampled correctly. However, following the request from the reviewer, we added the data for angiotensin IV. Interestingly, at pH 3.8 the translocation rate decreased and the blocked value is perfectly aligned with the predicted value (Fig. 5c).

- the "deviating" c-Myc peptide is the only negatively charged peptide at pH=4,5 ; it would have been of great interest to test other negatively charged peptides to see if the electrical charge is a critical parameter

We added another negatively charged peptide as suggested by the reviewer, which behaved as expected.

- what are the results of Fig 3 at pH=7,5 (more diversity in peptide charges than at pH=4,5)? use D109S-W116S FraC which provides the 3 nanopore types at pH=7,5 and 4,5 (see Supp Fig 2h and Supp Fig 3f) but does not seem to be exploited ; e.g. comparing type II nanopore analysis of the same peptides at pH=7,5 and at pH=4,5 could be of particular interest.

At pH 7.5 we did not observe the peptides entering the nanopore. We have now mentioned this in the main text.

Supp Fig 9: Angiotensin I is missing

we added the data for angiotensin IV

Fig 4:

need to be in very particular and strictly-controlled pH conditions (pH=3,8 ; >3,6 ; <4,0) for c-Myc to follow the Ires% vs peptide mass tendency of the other peptides

- b) what is the Ires% vs pH behaviour of the other peptides analyzed with type II nanopore (only 4 over 9 are shown)?

After we found that at pH 4.5 there is no correlation between the mass of the peptide and the current blockade, we tested other pH values. The rationale was to change the pH until all

peptides were positively charged. We found that only at pH 3.8 the four peptides showed a signal vs mass correlation. Most likely at pH 3.8 the constriction of FraC still carries a negative charge (the pore has seven protomers), while the peptides are largely positively charged. We then tested all the other peptides at this pH and found this correlation holds for all peptides. We did not perform the pH dependency for all peptides, because the correlation between $I_{ex\%}$ and the volume of the peptides is only found at pH 3.8.

- c) 2 peptides are missing

We added the two peptides missing.

- d) non monotonic behaviour of blockade duration of c-Myc vs pH for the different voltages ; maximum reached at pH=3,8 ; in addition to the need of strictly-controlled pH conditions, this suggests that pH=3,8 is a very particular pH value for this peptide, which creates doubt about the possible generalization of the technique ; testing other negatively charged peptides is of crucial importance as the main claim of the paper is the use of the system as a nanopore mass spectrometer

Following the reviewer request, we performed another experiment testing another negatively charged peptide. We found the new peptide (NLP-3) followed the predicted behavior. At pH 4.5 NLP-3 showed a higher Ires than other positive charged peptides, while at pH 3.8 NLP-3 Ires was in line with that of all the other peptides. Interestingly, since NLP-3 includes fewer acidic residues than c-Myc (the other negative peptide previously tested), NLP-3 showed a lower residual current at pH 4.5 than c-Myc. This suggests that at pH 4.5 the negative charge carried by both peptides might prevents them from reaching the sensing region within the nanopore.

Indeed, pH 3.8 is a “special case”. We think that, most likely, it is necessary to have a nanopore with a negatively charged constriction while the peptides should be positively charged (at low pH the N-terminus will always provide a positive charge, even for peptides with uncharged side chain).

Discussion:

lines 204-205: these are calculated values ; need to be measured experimentally

To avoid confusion, we have changed the text by saying that the value are just estimates.

lines 248-250: too much generalization regarding the results.

We have now specified that the pH is 3.8.

Reviewer #3 (Remarks to the Author):

In this manuscript, Huang et al. demonstrate FraC pores can be engineered to three sizes, with the relative yield of each size-type being tuneable by varying synthesis conditions. The authors then characterize mixtures of short (4 - 22 AA) peptides with improved sensitivity due to the longer dwell times and enhanced volumetric sensitivity elicited by the smaller (type II) pores. Methods to improve the yield of these smaller pores to appreciable percentage levels (e.g. from c.a. 0-3% to 14-20% for type III pores from WT-FraC) are presented. These include varying pH, the choice of chemical substitution at the lipid interface, elution, the concentration of monomers during liposomes and optimization of the purification protocol.

The authors then identify small volume differences (c.a. 40 Da) in mixtures of translocating peptides using the nanopores.

Importantly, under the right conditions, the signal appears proportional to the mass despite the fact that some substitutions have 'opposite charge'. Taken cumulatively, these findings are actually quite significant with respect to the literature; the production of sub-nm biological pores alone is novel. Being capable of identifying residue-scale volume differences in protein/peptides irrespective of charge is even more so. While there is a lot of pH dependent nanopore data in the literature, Figure 4 b and c are unusually striking examples of specificity. Overall, I recommend the paper for publication with minor corrections to be addressed below:

We are glad the reviewer recommended for publication and we appreciate the nice summary about our work.

General issues:

1. One issue is the question of robustness and longevity of the pores given the embedding protocol includes a (presumably weaker) lipid interface serine bond among other changes such as pH etc. Please provide a comment (or ideally quantitative data) on the lifetime of pores

broken out by type (I,II,III). Even if the types are equivalently functional, this would be instructive to the reader.

During the many measurements with these pores, we didn't observe any significant instability. All three type pores remained inserted into the lipid bilayer indefinitely. We have added this information in the main text.

2. Using pores with sub-nm or nearly nm cross section is expected to drastically increase clogging during translocation – a comment on the authors experience with the smaller pores would be useful. Potentially related to this, why were type II pores used predominantly for translocation experiments since 0.8 nm type I pores would give much more sensitivity to smaller peptides? Would this require too much bias to achieve translocation?

We understand the reviewer's concern, especially if the reviewer is used working with solid state nanopores. However, throughout the many measurements with our biological pores, we never observed any particular clogging during peptide translocation, even while using type III pores. Further, all types of pore reported in this paper showed no gating (e.g. reversible clogging).

We used mainly type II nanopores simply because they were more easily obtained. Further, type II nanopores allowed analyzing a wider range of peptide masses than type III nanopores. As for the bias required to translocate the peptides, we observed translocation using type III nanopores when we tested it.

3. It is not clear how many blockades are required in order to discriminate at 40 Da glutamate/alanine resolution (1,10,100 blockades?). A more statistically robust assertion of volumetric discrimination as a function of blockade count could strengthen the paper. A trend of 'volume sensitivity (Da)' plotted as a function of blockade count for the different pore types found using a boot strapping approach would add rigor to the claim of volume sensitivity. (One option would be to take N blockades, say 5 randomly chosen blockades of type alanine vs 5 others and find what Da resolution on average is achieved, then repeat for N = 1...1000 etc and plot against N - the blockade count). The trend should limit to say, 40 Da passed some sufficient number of molecules.

The reviewer has a good point. In fact, we are addressing this point in a separate manuscript. Here we only aimed at showing that two peptides differing by 44 Da can be visually separated in two groups. In further work we will show that two peptides showing smaller differences, which

cannot be visually separated, can be distinguished using a machine learning approach. In this work we would like to focus on the concept of nanopore mass identifier, and would like to give only a rough estimate about the resolution of this system.

4. It is also unclear how this work could be extended or enhanced - the authors commendably found small anomalies in their data (such as a <1% yield of 0.8 nm pores) and exploited it – but is there a way this could be extended to even smaller pores or is this a physical limit of the structure? Looking at the trend of smaller pore yield as a function of conditions, I initially wondered if the authors reduced the concentration of monomers during liposomes even further if yields could improve, but as in Figure S.1, a 1:80 protein:lipid ratio did not give nanopores in lipid bilayers. Did the authors ever observe even smaller pores. Is there an anticipated avenue for further engineering? Please add a discussion comment.

Occasionally, we observed yet smaller pores, however, they were too rare for a proper characterization. We have tried to engineer the surface of the nanopores with no success. We added in the main text a comment on the smaller nanopores.

5. The authors spend a lot of time saying mass spectrometry is limited in various ways "extremely complex and unwieldy, and are not single-molecule." The authors spend considerable time framing the work within the context of mass spectrometry so the point is salient, but I would argue that the 1 second acquisition required by a mass spectrometer to get a 0.001 Da accuracy mass identification outweighs the limits the authors put forward. There are actually a lot of synergies between mass spectrometry and nanopore sequencing. The nanopore mass spectrometer devices produced by Derek Stein are an excellent example of the potential complementarity between these technologies. The paper might benefit from a more complimentary introduction and discussion of the respective technologies.

We agree with the reviewer. We have now extended the introduction describing Derek Stein and other single-molecule approaches to protein recognition and sequencing.

Minor issues:

1. Claiming that the smaller pores were the 'dominant species' yield is somewhat unfair when both type II and III have to be grouped for this statement (line 108) – please rephrase (according to the supplement the increase was c.a. 0-3% to 14-20% for type III pores, but moreso in other

types – the reality is that results appear mixed and specific)

We rewrote this part.

2. In Figure 2, the authors have interestingly plotted I_{res} against S.D. because it can discriminate peptides – but why? Why does standard deviation permit distinction of species? Obviously translocation dynamics are the cause but in what sense? For example the angio II shows a larger standard deviation than angio I despite its smaller size. But then, angio IV (the smallest) inverts this trend by also having the smallest S.D current on average. Please explain or provide an intuitive explanation or justification for this in the text.

The reviewer identified an interesting effect. We do not know why blockades have different SD. Interestingly, this is observed in other biological nanopores and not only for FraC (e.g. Chavis, A. E. et al. ACS Sensors 2, 1319–1328 (2017)). We think the most likely possible interpretation to these data is that different peptides interact differently with the nanopore. Another possibility is that nanopore currents reported the conformational flexibility of the peptides.

3. Comparing Figure 2 to the state of the art, I note that the Wanunu lab “Identification of amino acids with sensitive nanoporous MoS₂: towards machine learning-based prediction npj 2D Materials and Applications (2018) 2:14” and Timp lab “Reading the primary structure of a protein with 0.07 nm³ resolution using a subnanometre-diameter pore - Nature nanotechnology 11 (11), 968 2016”, and Pevzner lab “Single-molecule protein identification by sub-nanopore sensors PLoS computational biology 13 (5), 2017” have separately found random forest regression to be the superior approach for classification of peptides blockades in sub-nanopores. Even if the authors do not want to perform a more complex analysis on their peptide blockades, I highly encourage these references be included in a statement that more complex classification of peptides has been demonstrated elsewhere, and would likely improve the sensitivity of discrimination - but S.D. and I_{res} are sufficient here for a proof of principle demonstration.

We have added a comment on the point raised by the reviewer and included the references.

4. The figure 2 surface plots require attention; please label the color bar axes, e.g. as ‘count’. Do the authors mean 0 to be white in the multi-colored label? Currently, ‘0’ is denoted a dark blue color in the color label, but no dark blue is present in the map despite 0 being present across most of the space. This is confusing when compared to Supplementary Figure 6 where

the background is actually blue – why? Contour plots are not as illustrative as kernel density maps for showing non-uniform distributions, but it's a superficial point but I encourage Figure 2, S6 and S7 to be redrawn.

We thank the reviewer for noticing this mistake. We have now re-drawn the figures by adding the contour plots for individual peptides, all having the same number of events. We think this representation will allow assessing more fairly the ability of the nanopore to address differences in peptide mixtures.

5. Continuing with Figure 2 iii-iv, the labels iii and iv are both used twice – is this intentional? If so, group the plots together and remove one label. The maps show different numbers of counts and different numbers in total – please normalize the map color height in some intuitive way (e.g. setting the max to 50, or to some fraction of total counts in each case).

We have changed the figure and used a fixed amount of data points. We also removed the labels.

6. It is hard to compare all 4 Figure 2 maps directly because the axes are not the same – please set the x axis (e.g. ranging from 0 to 45%) as constant in each case (instead of 2, 2). Consider logarithmic scales if it improves visualization and in the supplement.

We have changed the figure to show the separation of the peptides. For all peptides we used the same number of data points, except for the mixture, which sampled four peptides and contained four time more data points. Now each figure has the same scale.

7. 'Current blockade' can refer to the magnitude of current associated with the blockade of a pore, or to the absolute level of a pore when blocked. On lines 134 – 135, $I_{res}\%$, is defined as $IB / IO \times 100$, but if it was current magnitude then the residual current would be defined as $(1 - IB / IO) \times 100$ ". I recommend IB be more precisely defined as the ionic current level during a blockade.

We have defined IB as: the ionic current magnitude associated with a blockade. We also used excluded current defined as $IO - IB$ (see later). We used the term excluded current because it relates to the volume excluded by the peptide.

8. The same peptide will exhibit a different $I_{res}\%$ in different sized pores, because it will occupy

more/less volume. However, across those same pores, it will have same absolute current blockade magnitude provided the bias/electrolyte is constant. Therefore why is this statement true on line 134 that “Residual currents percent ($I_{res}\%$, defined as $I_B / I_O \times 100$) were used instead of current blockades because they provided more reliable values when comparing different nanopores.”? Wouldn't blockade magnitude, $|I_B - I_O|$, be superior across pores of different size because it is not sensitive to pore size?

We tried plotting our data as suggested by the reviewer but the correlation was not improved. This is most likely because most of the variation in nanopore recordings comes from small differences in open pore currents (the SD of the open pore current is about 5%), which in turn are likely to reflect differences in nanopore sizes. Normalizing by the open pore current compensate for most of the variation among nanopores.

9. In Figure 3, the residual current is broken out as a function of molecular weight. There is clearly a relationship between size and residual current. However, what is the purpose of performing a polynomial fit since the actual pore and peptide size are known as well as the translocation bias? Therefore the expected conductance, G can be explicitly determined and the peptide radii could be coarsely estimated as $V(\text{nm}^3) = 1.212 \times 10^{-3} (\text{nm}^3/\text{Da}) \times \text{MW}(\text{Da})$ as one example from which blockade fraction is easily estimated. An explicit fit or simple analytical expression using the known parameters should at least be attempted, as it would add further corroboration of the pore size if it fit the experimental data instead of just a guide to the eye. We thank the review for the suggestion. We have followed the suggestion and re-drawn the figure accordingly. This allowed us to use this correlation to estimate the volume of the recognition region in FraC.

10. It would improve the paper to add an analysis section to the methods and provide details like the filter settings in one place. The use of a 10 kHz is repeated in the Figure 1, Figure 2 and Figure 4 captions and again in table 1 and then again multiple times in the supplement. This seems wasteful since the vast majority of experiments were done in the same buffer at the same bias and settings. The manuscript would be improved if the authors gathered comprehensive details about how the data was handled, culled, cleaned, processed and plotted into the analysis section. While the filter used is relevant, it would be far more instructive to know how data was handled (manual cropping of current trace, one user/multiple users/repeatability?) and how specifically the tables were arrived at.

We have followed the reviewer suggestion and moved the experimental details to the methods section.

11. Please check the manuscript for typographical errors, e.g. “Bassel filter” should read “Bessel filter”

We have corrected this and other spelling mistakes to the best of our ability.

12. Supplementary Figure 9 caption – it is not amplitude it is residual blockade current fraction.

We have corrected this in Figure S9 and in other figure legends.

13. In supplementary Figure 8, Endomorphin I is badly fit since the Ires distribution is multimodal – is the secondary presence of many near 0% Ires measurements at long dwell times due to aggregates/clogging? The authors discussed this but do not attribute a cause or solution. Consider repeating the experiment with fresh, reduced analyte concentration. Figure. S11 appear to suggest this as being a multilevel blockade phenomena – if so please add a few raw events current traces showing multiple levels with level guidelines to Figure S11.

We added the traces in new Fig S13 and focused on separation of the two level events from other peptides. The most logical explanation for the bimodal distribution of events is that the events at 0.9% Ires are due to peptide dimers. Endomorphin-I (sequence: YPWF) contains several potential π - π interaction, as shown in the supramolecular assembly in aromatic di-and tri-peptides (e.g. Brown N, et al. ACS Nano, 2018, 12 (4), pp 3253–3262). Dilution of the sample (10 folds, from 4 μ M to 0.4 μ M), did not change the relative amount of the blockades, suggesting that the association constant of the peptides might be in the nM range.

Figure R1. Negative stain electron-microscopy investigation of W116S-FraC nanopores. The circles highlight the nanopores, which appear having different sizes.

Figure R2. Three repeats of angiotensin II (**a**) and angiotensin A (**b**) measurements using type II W116S-FraC pore at pH 4.5.

Figure R3. Plotting of angiotensin III and Ile7-angiotensin III mixture with different numbers of events considered.

REVIEWERS' COMMENTS:

Reviewer #1 (Remarks to the Author):

This is a heavily revised manuscript of earlier efforts. The authors have addressed many criticisms pertinent to Reviewer 1. Regardless, I would recommend taking "peptide/protein sequencing" out of the text for the simple reason that data from this manuscript does not support this claim. In its current formulation, this goal is still a distant dream.

Reviewer #2 (Remarks to the Author):

The authors addressed most of my comments and the manuscript now reads very well. The revision has improved the presentation of the manuscript. I recommend the manuscript to be published in Nature Communications. Please add these references in their context:
DOI: 10.1039/c6nr06936c
<https://doi.org/10.1038/nbt.4278>

Reviewer #1 (Remarks to the Author):

This is a heavily revised manuscript of earlier efforts. The authors have addressed many criticisms pertinent to Reviewer 1. Regardless, I would recommend taking "peptide/protein sequencing" out of the text for the simple reason that data from this manuscript does not support this claim. In its current formulation, this goal is still a distant dream.

This work shows for the first time that nanopores can be used as a single-molecule mass spectrometer for peptides, and mass-spectrometry is the default technique to sequence proteins. However, we agree a protein sequencing device has not been obtained yet. Hence, we made a series of changes to tone down the link with protein sequencing. The rationale of the changes is to better emphasize the steps towards the final goal of single-molecule protein sequencing.

In particular, we have updated the reference to protein sequencing in the abstract emphasizing peptide-mass identification at the single-molecule level. In the introduction we have left the reference on protein sequencing as this is the final aim. However, we have also added "protein and peptide identification", as this describes more closely what has been achieved by our work. We also specified at the end of the abstract that this is one crucial step towards single-molecule protein sequencing.

Reviewer #2 (Remarks to the Author):

The authors addressed most of my comments and the manuscript now reads very well. The revision has improved the presentation of the manuscript. I recommend the manuscript to be published in Nature Communications.

Please add these references in their context:

DOI: 10.1039/c6nr06936c

<https://doi.org/10.1038/nbt.4278>

We are glad the reviewer was satisfied with the additions. We have added the references.